# Quantum suppression of cold reactions far from the *s*-wave energy limit

Or Katz [1] ✉, Meirav Pinkas [2], Nitzan Akerman[2], Ming Li[3,4] & Roee Ozeri[2]

Quantum effects in chemical reactions are most pronounced at ultracold temperatures, where only a few partial waves contribute. While interference among many partial waves is theoretically expected to persist at higher temperatures, direct evidence for such quantum effects in reactive processes has been lacking. Here, we report signatures of quantum interference suppressing a chemical reaction in the multi-partial-wave regime: resonant charge exchange between a single $^{87}Rb^+$ ion and its parent atom $^{87}Rb$. Using quantum-logic detection on a single atom-ion pair and a calibrated in-situ measurement of Langevin collision probabilities, we benchmark the thermally averaged reaction rate against both classical and quantum predictions. We find that the reaction rate is suppressed by over an order of magnitude relative to the classical expectation, despite occurring in the millikelvin temperature regime (more than three orders of magnitude above the *s*-wave threshold), where more than a dozen partial waves contribute. These results suggest quantum interference as a key mechanism in chemical reactivity beyond the ultracold limit and offer a platform for probing coherent quantum effects in atom-ion reactions where ab initio methods remain intractable.

Do quantum interference effects persist in chemical reactions far outside the *s*-wave regime? At ultracold temperatures, where the de Broglie wavelength of atoms exceeds their interaction range, the wave-like nature of matter often governs collision and reaction dynamics. In this regime, quantum phenomena such as shape[1] and Feshbach resonances[2–4] are especially pronounced and tunable; though such resonances can also occur at higher energies and higher partial waves in crossed-beam and related experiments[5–7]. These effects underpin advances in quantum simulation[8–10], ultracold chemistry[11], and quantum sensing[12,13].

Outside this regime, scattering typically involves many angular momentum channels, or partial waves, each labeled by a quantum number *l*. For neutral atoms, the ultracold *s*-wave regime (*l* = 0) is reached only below a temperature of a few hundred microkelvin[11,14–19], whereas at higher energies, the centrifugal and van der Waals inter-action potentials differ across partial waves, accumulating distinct scattering phase shifts. Thermal averaging over many partial waves

typically suppresses interference effects, even when individual reso-nances are present, leading to classical behavior. In such systems, observing quantum interference effects typically requires ultracold temperatures (e.g.,[20–22]) or is limited to special cases such as collisions between identical particles[23,24].

Collisions between neutral atoms and ions differ significantly from neutral-neutral interactions due to the long-range polarization potential that governs atom-ion dynamics. This long-range interaction reduces the energy scale for entering the *s*-wave regime to sub-$\mu$K for most atom-ion pairs[17,18,25–29], and has motivated extensive experimental studies of atom-ion collisions across a wide range of energies[25,30–51]. However, in ion traps, residual micromotion and other trap-induced heating effects pose significant challenges to reaching this ultracold regime[25,27,28]. As a result, most hybrid atom-ion experiments operate at higher temperatures where many partial waves contribute, limiting access to interference effects in elastic observables such as scattering lengths. In contrast, inelastic collisions and chemical reactions are

[1]School of Applied and Engineering Physics, Cornell University, Ithaca, NY, USA. [2]Department of Physics of Complex Systems, Weizmann Institute of Science, Rehovot, Israel. [3]Atom Computing, Inc., 2500 55th St, Suite 100, Boulder, Colorado, USA. [4]Present address: Google Quantum AI, Venice, CA, USA. ✉e-mail: or.katz@cornell.edu

predominantly governed by short-range molecular dynamics, where coupling between atomic wavefunctions becomes substantial and relative scattering phase shifts across relevant channels vary more uniformly with partial waves. This opens the possibility for quantum interference effects to persist even in the multi-partial-wave regime and be resillient to thermal averaging. Such behavior was proposed for inelastic collisions[52] and for chemical reactions[53,54], through a mechanism known as partial-wave phase locking. When the scattering phase shifts of many contributing partial waves are nearly aligned with that of the s-wave, coherent interference can survive far beyond the ultracold limit, inducing s-wave-like behavior in thermally averaged reaction and inelastic rates even at millikelvin temperatures.

Preliminary signatures of partial-wave phase locking in inelastic spin-changing processes were previously reported in our studies of spin-exchange collisions between Sr$^+$ ions and neutral $^{87}$Rb atoms[51,52,55–58]. To enable such measurements across different isotopes, we developed a quantum-logic detection technique that allowed us to probe multiple ion species within the same experimental apparatus[56], and identified a trap-induced dynamical mechanism that systematically affects observed cross sections[57]. Analysis of multi-isotope, multi-channel data revealed behavior consistent with partial-wave phase locking in inelastic spin-changing channels[58]. While these results revealed deviations from classical predictions, much of the observed suppression in spin-exchange cross sections was largely driven by isotope-dependent variations in Sr$^+$ across isotopes, and the overall effect across spin channels associated with phase-locking was limited. These studies established a framework for quantum control and interpretation of atom-ion collisions at elevated temperatures, but left a key open question: can partial-wave phase locking manifest in a chemical reaction, such as charge exchange, that transforms one atomic species into another, leading to a substantial deviation from classical behavior?

Here, we demonstrate that quantum interference can substantially suppress a chemical reaction far from the s-wave limit. We study resonant charge exchange between ultracold $^{87}$Rb atoms and a single cold $^{87}$Rb$^+$ ion, a binary chemical reaction involving identical nuclei at temperatures more than three orders of magnitude above the s-wave threshold. Although $^{87}$Rb$^+$ lacks accessible optical transitions for direct detection, we probe its dynamics using a quantum-logic protocol with a co-trapped $^{88}$Sr$^+$ ion[56]. To quantify reaction probabilities, we introduce and validate an experimental method for extracting Langevin collision rates from momentum-changing events. We compare the results with multichannel quantum defect theory (MQDT), finding strong agreement with quantum predictions. The measured rate is suppressed by more than an order of magnitude relative to the classical expectation. This confirms that coherent interference among many partial waves can determine reactive outcomes even in the millikelvin regime. Our results provide signatures of partial-wave phase locking in a chemical reaction and establish a broadly applicable framework for exploring quantum effects in reactions beyond the reach of classical theory.

## Results

### Interference pathways in resonant charge exchange

Resonant charge exchange between an ion and its parent atom offers a pristine setting to explore quantum interference in chemical reactions. In particular, phase-locking mechanism predicted in ref. 53 for chemical reactions suggests that quantum interference may persist under near-resonant conditions, where the binding energy of the reactants is nearly identical to that of the products, a condition naturally realized in ion-parent-atom systems. For alkali-metal species, this process involves a ground-state neutral atom in a $^2S_{1/2}$ state and a singly charged ion in a $^1S_0$ state. At large separations, the atom-ion interaction is dominated by an attractive polarization potential, $V(R) = -C_4/R^4$, where $C_4$ is the atom-ion polarization coefficient depending on the

static polarizability of the neutral atom[25]. In a classical picture, this leads to inward-spiraling Langevin trajectories when the centrifugal barrier is overcome[25].

At short range, where electronic wavefunctions overlap, the system is described by two electronic Born-Oppenheimer molecular potentials: a symmetric (gerade) and an antisymmetric (ungerade) state. These define two quantum pathways for electron transfer, and their coherent interference governs the reaction outcome. In the absence of spin, the cross section for resonant charge exchange is given by:

$$\sigma_{\mathrm{RCE}} = \frac{\pi\hbar^2}{2\mu E}\sum_l (2l+1)\sin^2(\delta_g^{(l)} - \delta_u^{(l)}), \tag{1}$$

where $\delta_g^{(l)}$ and $\delta_u^{(l)}$ are the scattering phase shifts for the gerade and ungerade potentials in each partial wave $l$ and energy $E$, and $\mu$ is the reduced mass. This expression sums the contributions from many partial waves. The classical, high-temperature limit is reached when the summation is effectively incoherent because the phase difference $\delta_g^{(l)} - \delta_u^{(l)}$ varies significantly with $l$. However, if this relative phase difference depends weakly on $l$, the interference factor is similar across many partial waves, so the thermal average retains s-wave-like behavior at elevated energies[53]. In this regime, the reaction rate can be either suppressed or enhanced relative to the classical value.

Direct detection of resonant charge exchange in cold ion-parent-atom systems is nontrivial. Although the atom and ion differ by their valence electron, the reactants and products are indistinguishable in both mass and net charge. Moreover, low-energy scattering is nearly isotropic. As a result, momentum-based detection provides no clear signature of reaction events[59]. To overcome this, we exploit the intrinsic hyperfine structure of alkali-metal atoms, using the nuclear spin to distinguish initial and final states. Specifically, we consider $^{87}$Rb$^+$ ions and their parent neutral atoms, both of which carry nuclear spin $I = 3/2$ and can begin in distinct nuclear spin states. The neutral atom exhibits ground-state hyperfine structure with total angular momentum $F = 1$ and $F = 2$, while the closed-shell $^{87}$Rb$^+$ ion has no hyperfine structure, as shown in Fig. 1.

This internal structure enables us to probe charge exchange reactions through a mechanism analogous to a two-path molecular interferometer. At large separations, the atomic hyperfine state defines a well-characterized entrance channel. As the particles approach, electronic and hyperfine interactions couple non-perturbatively, and the hyperfine basis no longer diagonalizes the system. The wavefunction evolves into a superposition of short-range molecular channels, gerade and ungerade, which acquire a relative phase during the evolution, similar to the arms of an interferometer. Upon separation, the system projects back onto the hyperfine basis, and interference between outgoing amplitudes can redistribute population across final hyperfine states, for instance resulting in a hyperfine transition from $F = 2$ to $F = 1$ of the neutral atom. We refer to this particular spin-resolved process as hyperfine de-excitation resonant charge exchange (HDRCE) as shown in Fig. 1a for the s-wave limit and Fig. 1c for the finite temperature case. While HDRCE modeling requires a multichannel extension of Eq. (1), it provides a spin-selective probe of quantum interference in resonant charge exchange reaction, ideally suited to cold homonuclear systems where phase-locking is predicted[53] but conventional detection is insensitive.

While the framework of partial-wave phase locking suggests that interference may persist in resonant reactions[53], its occurrence in specific systems is not reliably predictable from theory. In particular, for $^{87}$Rb – $^{87}$Rb$^+$, the magnitude of interference effects depends on short-range scattering phase shifts that cannot currently be accurately computed from first principles using ab initio calculations. As a result, it remained unknown whether this reaction would exhibit

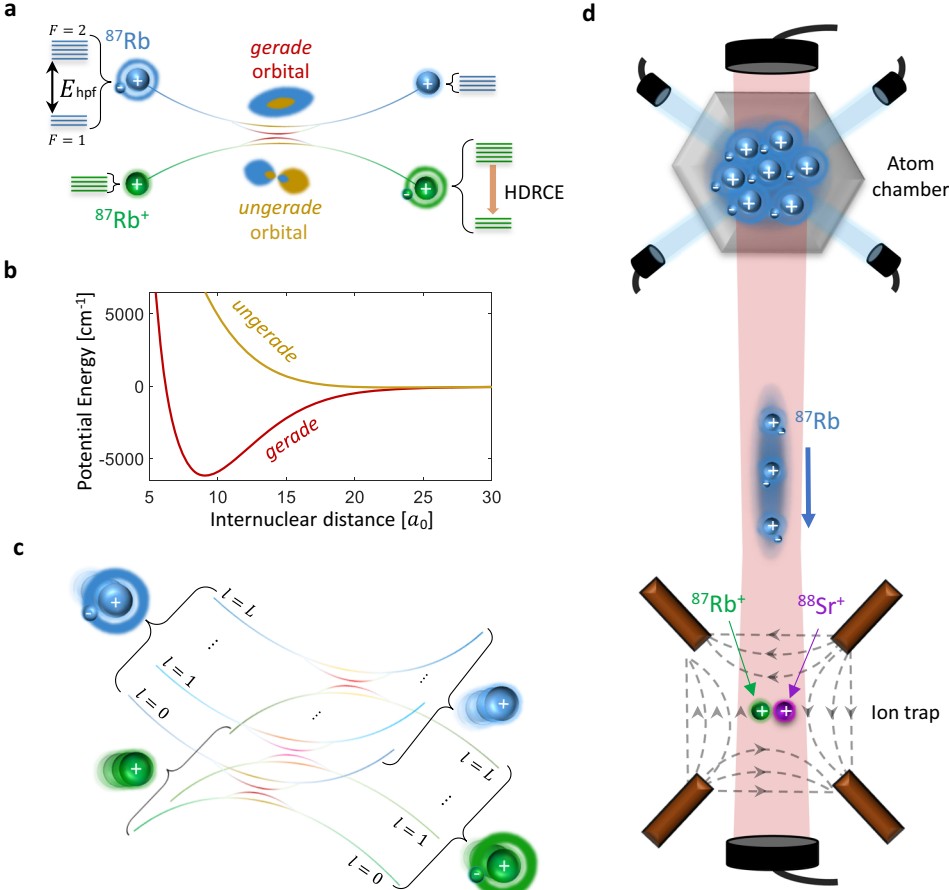

**Fig. 1 | Quantum interference in atom-ion charge exchange reactions. a** At ultracold temperatures, where only a single partial wave contributes, resonant charge exchange between an ion and its parent neutral atom proceeds via coherent interference between two molecular channels: the symmetric (gerade) and antisymmetric (ungerade) electron configurations. The reaction acts as a two-path interferometer, with the reaction probability determined by the relative scattering phase between the two channels. For $^{87}Rb - {}^{87}Rb^+$, both species carry nuclear spin $I = 3/2$, enabling spin-resolved identification of the reaction via hyperfine de-excitation of the neutral atom (HDRCE). $E_{hpf}$ denotes the hyperfine energy and $F$ the spin quantum number. **b** Molecular potential energy curves for the gerade and ungerade channels in the s-wave limit. Short-range energy differences between the two curves accumulate a relative scattering phase that determines the interference outcome and governs the reaction probability (see Eq. (1)). These phases are highly sensitive to molecular details and remain beyond

current ab initio predictive capability, limiting theoretical estimates of the reaction rate. $l$ denotes the partial wave. **c** At elevated temperature, many partial waves contribute to the reaction. In the classical regime, their contributions are summed incoherently, and thermal averaging tends to wash out quantum interference. If, however, the short-range phase difference between the gerade and ungerade channels changes only weakly with $l$ and with collision energy, the interference factor remains nearly the same across many partial-wave terms. In this case, averaging over partial-wave contributions can still preserve wave-specific interference signatures beyond the s-wave limit, a phenomenon known as partial-wave phase locking[53]. **d** Schematic of the experimental platform. A cloud of ultracold $^{87}Rb$ atoms is shuttled into an ion trap containing a two-ion crystal composed of $^{87}Rb^+$ and $^{88}Sr^+$. Individual resonant charge exchange events are detected via energy released into the ion crystal and read out using quantum logic techniques on the $^{88}Sr^+$ logic ion.

enhancement, suppression, or near-classical behavior. Our measurements experimentally answer this question.

## Experimental study

We investigate hyperfine de-excitation resonant charge exchange (HDRCE), a process in which a neutral atom undergoes a hyperfine transition during resonant charge exchange with its parent ion. The experiment uses a hybrid atom-ion platform capable of detecting rare scattering events with spin and energy resolution. A laser-cooled cloud of $^{87}Rb$ atoms is initially confined in an optical lattice and adiabatically transported into a neighboring vacuum chamber containing a two-ion crystal in a linear Paul trap (Fig. 1d). The crystal comprises one $^{87}Rb^+$ ion and one $^{88}Sr^+$ ion. Since $^{87}Rb^+$ has a closed electronic shell and no optically accessible transitions, direct detection and control are infeasible. Instead, the co-trapped $^{88}Sr^+$ ion serves an auxiliary role, providing cooling, calibration, and detection. In each run, the neutral $^{87}Rb$ atoms are optically pumped into one of the five Zeeman sublevels

of the $F = 2$ hyperfine manifold. The $^{88}Sr^+$ ion is prepared in the spin-up state, and the $^{87}Rb^+$ ion is unpolarized (see Methods). The atom density is tuned such that, the average number of atom-ion collisions per cloud passage through the ion trap is below one.

HDRCE reactions are detected through their associated energy release: when a neutral $^{87}Rb$ atom transitions from $F = 2$ to $F = 1$, it releases the internal hyperfine energy. In principle, the reaction could be identified by directly measuring the final spin state of the neutral atom. However, in practice, the atom is untrapped after the collision, and interrogating its state is experimentally unfeasible. Instead, we detect such reactions via motional excitation of the ion crystal, which is mapped onto the internal state of a co-trapped logic ion and read out using carrier-shelving thermometry[56]. While this method was originally developed for inelastic spin-exchange collisions[56], here we adapt it to probe rare chemical reactions. The detection is efficient for hyperfine-level energy release and robust to background heating at the milli-kelvin scale (see "Methods").

Neutral $^{87}$Rb atoms can collide with either ion in the crystal. In $^{87}$Rb – $^{87}$Rb$^+$ collisions, hyperfine de-excitation occurs via resonant charge exchange, where the atom and ion exchange an electron, effectively swapping their roles as atom and ion. Other spin-changing processes are strongly suppressed (see "Methods"). In contrast, $^{87}$Rb – $^{88}$Sr$^+$ collisions involve spin exchange between valence electrons without charge transfer, with minor contributions from spin relaxation processes[52]. For simplicity, we refer to all hyperfine-changing processes in the Rb − Sr$^+$ system as spin exchange, though our analysis does not depend on the microscopic spin-changing pathway. For atoms initially prepared in the $F = 2$ such collisions can induce hyperfine de-excitation, producing a measurable increase in the motional excitation probability per cloud passage (see Fig. 2). Under our operating magnetic field ($B = 3$ G), changes in Zeeman energy are negligible compared to the hyperfine splitting, so the net energy release is effectively set by the hyperfine transition.

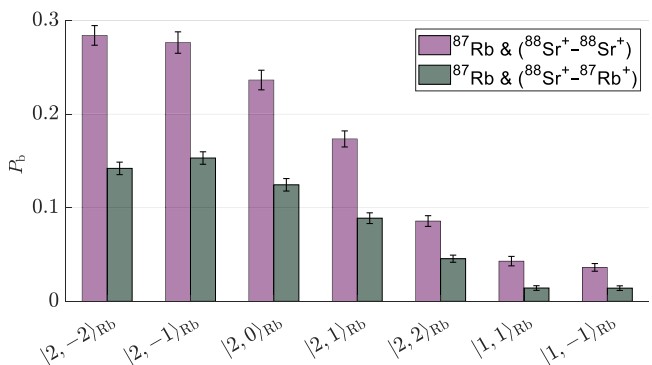

**Fig. 2 | Hyperfine de-excitation probability.** Measured heating probabilities following a single passage of the atomic cloud through the ion trap in a two-ion crystal. Data are shown for two configurations: a $^{87}$Rb$^+$ – $^{88}$Sr$^+$ crystal (gray) and a control $^{88}$Sr$^+$ – $^{88}$Sr$^+$ crystal (purple). The $^{87}$Rb atoms are prepared in a defined $|F, M\rangle$ spin state (horizontal axis). Heating is detected via carrier shelving thermometry, where the probability $P_b$ corresponds to detecting at least one bright $^{88}$Sr$^+$ ion, indicating significant motional excitation. For $F = 2$, this primarily reflects hyperfine-changing collisions; for $F = 1$, it indicates the background false detection rate. Error bars denote $1\sigma$ binomial uncertainties.

To isolate the HDRCE contribution, we perform reference measurements using $^{88}$Sr$^+$ – $^{88}$Sr$^+$ crystals, where neutral $^{87}$Rb atoms undergo spin exchange with either logic ion, where both ions can participate in spin exchange. Control experiments with atoms in the $F = 1$ manifold suppress transitions that release hyperfine energy and establish a background level (see Methods). From these measurements, we extract the HDRCE probability per colliding $^{87}$Rb – $^{87}$Rb$^+$ pair and find it to be nearly an order of magnitude smaller than the hyperfine de-excitation probability in spin-exchange collisions with a $^{87}$Rb – $^{88}$Sr$^+$ pair (Fig. 3).

The probability for an atom and ion to reach short range is set by their long-range interaction, which depends primarily on the atom-ion reduced mass and the neutral atom polarizability. These parameters are nearly identical for Rb$^+$ and Sr$^+$, so the Langevin collision probability for the two systems is effectively the same. Since the spin-exchange process itself occurs at a rate below the Langevin limit[58], this indicates that the observed HDRCE rate is strongly suppressed relative to the total Langevin capture rate. However, while the classical rate for resonant charge exchange is expected to be approximately half of the Langevin rate[26], our HDRCE measurement reflects only a subset of charge-exchange events, specifically those accompanied by a hyperfine transition. This spin-selective nature introduces statistical prefactors that reduce the measured HDRCE rate independently of the underlying charge-exchange probability. To distinguish these spin-related reductions from non-classical suppression of charge exchange, we perform an in-situ calibration of the total Langevin collision rate and compare the results with a multi-channel theoretical model that quantifies the spin-statistical contributions. This combined approach enables a direct and quantitative comparison between our measurements and theoretical predictions.

We first extract an absolute rate coefficient from the observed HDRCE probability. Although this conversion generally depends on experimental parameters such as atom density[18,25,30,39,52], the geometry and speed of the atomic cloud[55], and ion trap characteristics that may influence trap-induced dynamics and quasi-bound state formation[57,60], we circumvent these complications by directly calibrating the total Langevin collision probability in situ. We measure the rate of momentum-changing collisions under controlled excess micromotion (Figs. 3b and 4a) to obtain a reliable estimate of the Langevin rate in our experimental setting. This calibration allows us to extract the

**a**

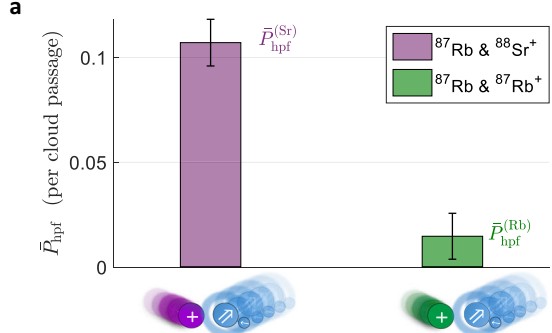

**b**

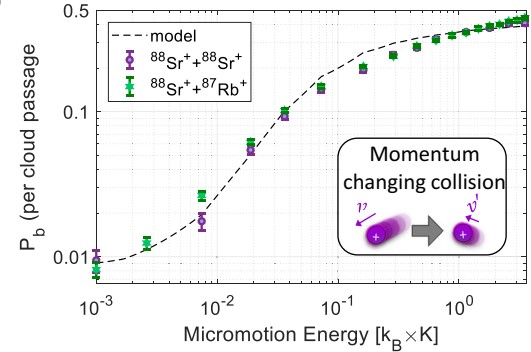

**Fig. 3 | Detection of hyperfine de-excitation and calibration of momentum-changing collision rate. a** Probabilities for hyperfine de-excitation of a $^{87}$Rb atom $P_{hpf}$ during a single passage of the atomic cloud through the ion trap, due to either resonant charge exchange reactions with a $^{87}$Rb$^+$ ion (green) or spin exchange collisions with a $^{88}$Sr$^+$ ion (purple). These events are identified via energy deposited into the ion crystal and read out using a quantum logic protocol (see "Methods"). Each value is averaged over initial Zeeman sub-levels in the $F = 2$ manifold of $^{87}$Rb. **b** Probability of momentum-changing collisions per cloud passage, measured by inducing controlled excess micromotion and using atoms prepared in the $F = 1$ manifold, where hyperfine-changing

transitions are suppressed. These events convert micromotion energy into secular motion and enable a robust, in-situ calibration of the Langevin collision rate using numerical simulations. The observed increase in $P_b$ with micromotion arises from larger momentum transfer into secular motion during collisions, which can cause shelving failures into the $D$ orbital and result in a higher probability of detecting a bright ion. This calibration method, introduced in this work, enables conversion of observed reaction probabilities into absolute rate coefficients for comparison with theory (see text and Methods). Dashed lines indicate predictions from numerical simulations. Error bars represent one standard deviation ($1\sigma$) binomial uncertainties.

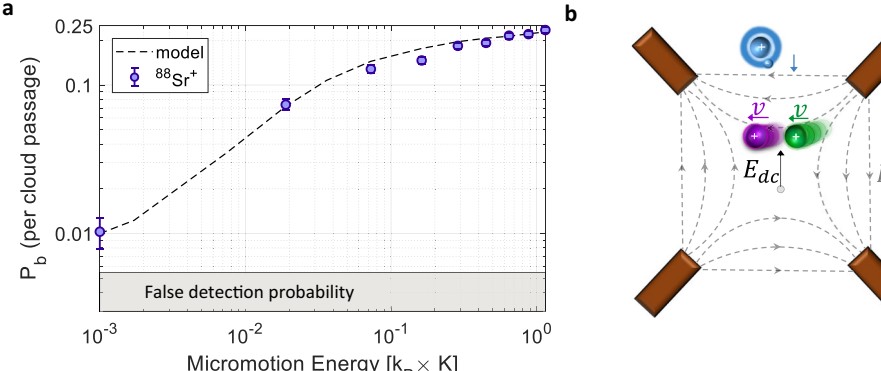

**Fig. 4 | Momentum-changing collisions.** These collisions convert the micromotion energy of an ion into secular motion, detectable with probability $P_b$ per passage of the atomic cloud. **a** Measured heating probability for a single-ion crystal of $^{88}Sr^+$. Dashed lines show numerical simulations, indicating an average Langevin collision rate per ion of $\kappa_L = 0.29 \pm 0.02$ per cloud passage. Error bars represent $1\sigma$ binomial uncertainties. **b** For measurements of hyperfine-changing collisions, excess micromotion is actively compensated. In contrast, to study momentum-changing collisions, a static electric field $E_{dc}$ is applied during the experiment to displace ions from the RF null, introducing excess micromotion along the RF field lines ($E_{RF}$). The illustrated ion crystal corresponds to the two-ion configuration used in Fig. 3b.

HDRCE rate coefficient and directly compare it with theoretical predictions for resonant charge exchange, as detailed in the Methods. We find the HDRCE rate coefficient is $k_{HDRCE} = (0.015 \pm 0.012) \times k_L$, where $k_L = 2.45 \times 10^{-9}$ cm$^3$/s is the classical Langevin rate coefficient for $^{87}Rb^+ - ^{87}Rb$ collisions in free space and theoretically given by the energy-independent expression[61] $k_L = 2\pi\sqrt{C_4/\mu}$, where $\mu$ is the reduced atom-ion mass.

## Theoretical Framework

We interpret the observed rate of hyperfine de-excitation resonant charge exchange (HDRCE) using multichannel quantum defect theory (MQDT)[62,63], which provides a framework to quantify quantum interference effects in reaction rates within the multi-partial-wave regime. In contrast to previous theoretical treatments that included only long-range polarization forces and short-range scattering phase shifts[53], our analysis also incorporates hyperfine interactions and the full multichannel structure necessary to describe hyperfine-changing processes.

In MQDT, the complex short-range dynamics are captured by four energy- and partial-wave-independent parameters: two quantum defects, $\mu_g^{(0)}, \mu_u^{(0)}$, which describe short-range phase shifts for the gerade and ungerade molecular potentials in the $s$-wave and zero-energy limit, and two curvature parameters, $\beta_g, \beta_u$, which approximate the angular-momentum dependence of the short-range phase shifts. While $\beta_g$ and $\beta_u$ can be reliably estimated using semiclassical WKB methods applied to ab initio potential energy surfaces, the absolute values of $\mu_g^{(0)}$ and $\mu_u^{(0)}$ are highly sensitive to short-range molecular details and cannot be predicted theoretically (see "Methods"). This highlights a fundamental limitation of ab initio approaches in cold atom-ion systems: the direction and magnitude of interference effects affecting reaction rates, which depend primarily on the difference of quantum defects $\mu_g^{(0)} - \mu_u^{(0)}$, remain unknown without experimental input.

To compare with experiment, we numerically compute thermally averaged HDRCE rate coefficients, which integrate the numerically calculated energy-dependent cross section over the relative velocity distribution at a given temperature. At the experimental temperature of approximately 0.6 mK, about a dozen partial waves contribute significantly. Nevertheless, our estimates and MQDT model confirm that coherent interference can indeed persist under these conditions. As shown in Fig. 5a, the rate remains nearly constant with temperature across different levels of approximations. The full model includes $\beta_g$ and $\beta_u$ values computed from ab initio potentials for the $^{87}Rb - ^{87}Rb^+$ system. This behavior provides theoretical confirmation of the partial-wave phase-locking mechanism, showing that the thermally averaged reaction rate remains approximately constant well beyond the ultracold regime[53].

Figure 5 also illustrates how different values of $\mu_g^{(0)}$ and $\mu_u^{(0)}$, the theoretically unconstrained quantum defects, can yield reaction rates that range from strong suppression to enhancement relative to the classical prediction (black line). Curves of the same color correspond to same parameters but different MQDT approximations, isolating the role of angular momentum dependence, as described in the Methods section.

To provide physical intuition, we highlight two extreme theoretical limits. In the fully phase-locked limit, where curvature parameters vanish ($\beta_g = \beta_u = 0$), the thermally averaged rate takes the form:

$$k_{HDRCE} = \frac{I}{2I+1} \sin^2\left[\pi(\mu_g^{(0)} - \mu_u^{(0)})\right] k_L. \qquad (2)$$

The HDRCE rate is minimized when $\mu_g^{(0)} = \mu_u^{(0)}$ and maximized when the two differ by one-half. In contrast, in the classical limit where short-range phases vary randomly with $l$ and all partial-wave contributions add incoherently, the HDRCE rate converges to a constant fraction of the Langevin rate

$$k_{HDRCE} \rightarrow \frac{I}{4I+2} k_L, \qquad (3)$$

as predicted by the degenerate internal state approximation (DISA)[64,65], corresponding to 3/16 of the Langevin rate for $^{87}Rb - ^{87}Rb^+$. This classical value serves as a benchmark for comparison with the observed suppression. A full account of the MQDT model, including parameter estimation and the numerical methods we use are detailed in the Methods.

## Hyperfine de-excitation resonant charge exchange

Figure 5a compares the measured HDRCE rate coefficient (black point) with theoretical predictions. Each color corresponds to a different value of quantum defects $\mu_g^{(0)}, \mu_u^{(0)}$, while line styles distinguish between full MQDT calculations and semi-classical approximations. In all cases, the rate remains nearly constant across the millikelvin regime, indicating the persistence of partial-wave phase locking.

The measured rate lies approximately twelve times below the classical prediction (black dash-dotted line in Fig. 5a), which corresponds to incoherent partial-wave summation valid in the high-temperature limit. This suppression is captured by the green curves in

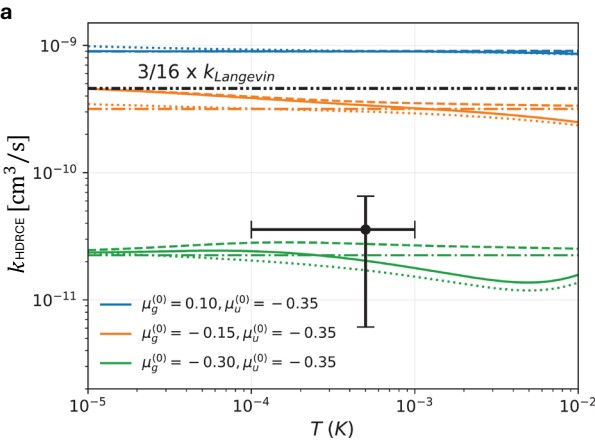

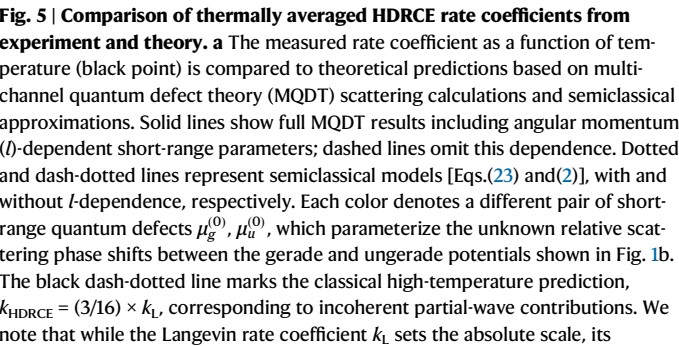

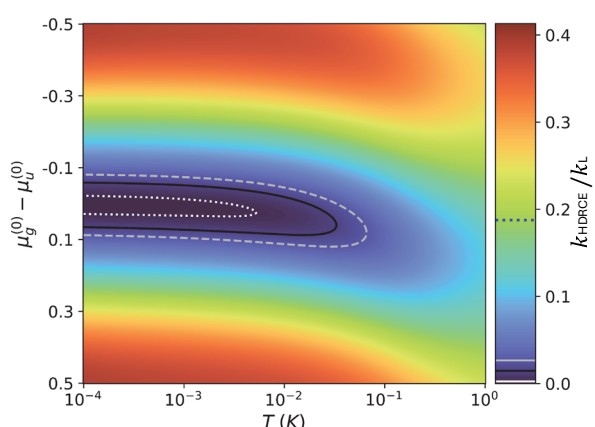

**Fig. 5 | Comparison of thermally averaged HDRCE rate coefficients from experiment and theory. a** The measured rate coefficient as a function of temperature (black point) is compared to theoretical predictions based on multi-channel quantum defect theory (MQDT) scattering calculations and semiclassical approximations. Solid lines show full MQDT results including angular momentum ($l$)-dependent short-range parameters; dashed lines omit this dependence. Dotted and dash-dotted lines represent semiclassical models [Eqs.(23) and (2)], with and without $l$-dependence, respectively. Each color denotes a different pair of short-range quantum defects $\mu_g^{(0)}$, $\mu_u^{(0)}$, which parameterize the unknown relative scattering phase shifts between the gerade and ungerade potentials shown in Fig. 1b. The black dash-dotted line marks the classical high-temperature prediction, $k_{\mathrm{HDRCE}} = (3/16) \times k_L$, corresponding to incoherent partial-wave contributions. We note that while the Langevin rate coefficient $k_L$ sets the absolute scale, its

theoretical value only uniformly rescales both the computed and experimental results. Vertical error bars represent one-standard-deviation ($1\sigma$) binomial uncertainties, and the horizontal bar indicates the systematic range bounded by our model fit. **b** Contour map of the ratio $k_{\mathrm{HDRCE}}/k_L$ as a function of temperature and quantum defect difference $\mu_g^{(0)} - \mu_u^{(0)}$, computed using the semiclassical model in Eq. (23) with phase shifts derived from Eq. (24). The solid black contour shows the locus of theory points consistent with the central experimental value while the dashed gray and dotted white contours indicate the corresponding $\pm 1\sigma$ bounds. Suppression below the classical limit (blue dashed line on the color scale) occurs only within a narrow region of near-equal short-range phases, consistent with partial-wave phase locking, extending a large temperature range relative to the $s$-wave limit (~$0.08\mu$K). At higher temperatures ($T \gtrsim 0.05$K), the model predicts that suppression weakens as partial-wave phase locking breaks down.

Fig. 5a, corresponding to a small quantum defect difference $\mu_g^{(0)} - \mu_u^{(0)} = 0.05$, indicating that short-range phase shifts remain nearly aligned across partial waves contributing at the experimental temperature. The observed deviation from the classical limit thus constrains the difference between the $s$-wave quantum defects, constraining the difference in ultracold scattering lengths despite the elevated temperature.

To examine the robustness of this suppression across temperature, Fig. 5b presents a contour map of the thermally averaged HDRCE rate coefficient as a function of temperature and $\mu_g^{(0)} - \mu_u^{(0)}$, calculated using the semi-classical model from Eq. (23) (see "Methods" details). The solid, dashed, and dotted contours shows the locus of theory points consistent with the central experimental value and its uncertainty bounds. The measured rate is reproduced only within a narrow range of small quantum defect differences and remains approximately constant up to tens of millikelvin. Beyond $T \sim 0.05$ K, the growing variation in short-range phase shifts across partial waves weakens interference, and by $T \gtrsim 1$ K, the predicted rate converges toward the classical limit.

## Discussion

The observed suppression of resonant charge-exchange reactions provides signatures of coherent quantum interference persisting far above the $s$-wave regime. The manifestation of the phase-locking mechanism in heavy atom-ion systems such as $^{87}$Rb − $^{87}$Rb$^+$ cannot be computed reliably, limiting ab initio predictions of quantum effects on classically expected outcomes. Our measurements thus provide partial information on the scattering lengths between the gerade and ungerade molecular potentials in the $s$-wave limit, based on data acquired at temperatures over three orders of magnitude above the $s$-wave threshold of $0.08\,\mu$K. Specifically, they constrain the absolute value of the relative difference between the short-range quantum defects to be small compared to unity.

Other reaction pathways, including spin relaxation and three-body processes, are energetically forbidden or strongly suppressed

under our experimental conditions. Trap-induced dynamics, which can enhance reaction rates by increasing the frequency of short-range encounters under classical motion[57,60], are explicitly included in our analysis. The measured suppression is therefore most consistent with arising from reduction in the resonant charge-exchange rate coefficient.

The sinusoidal dependence of the reaction rate on the short-range phase difference (Eq. (2)) suggests that other atom-ion systems may exhibit either suppression or enhancement of their classical value, depending on molecular details. Theoretical calculations of lighter species such as hydrogen[63], lithium[66] or beryllium[67], which are more theoretically tractable due to fewer electrons, predict deviations from classical calculations, although these predictions remain experimentally unverified. In contrast, charge exchange between heavier elements[68] such as Yb$^+$ experiments report reaction rates near or above the Langevin limit[30], but may be influenced by trap-induced enhancements[57] and require further analysis to disentangle trap effects from intrinsic reaction dynamics.

Looking ahead, improved trapping and detection methods could enable energy-resolved measurements of charge-exchange reaction rates across a broader range of collision energies and temperatures. Such measurements would allow us to trace the predicted transition to the classical regime, where the HDRCE rate is expected to increase with collision energy. Approaches such as optical trapping or temporarily switching off trap fields during collisions would lower the minimum achievable temperature and complement the thermally averaged results presented here. Furthermore, studying charge exchange between different isotopes, such as $^{85}$Rb − $^{85}$Rb$^+$ or other elements, is a compelling direction that could reveal how changes in reduced mass and hyperfine structure influence phase-locking and suppression, providing a stringent test of the theoretical framework. Finally, applying modest magnetic fields (on the order of tens of gauss) could shift the populations of different Zeeman spin states. With appropriate adjustments to the logic detection scheme, this could open the door to spin-selective studies of resonant exchange processes, or enable

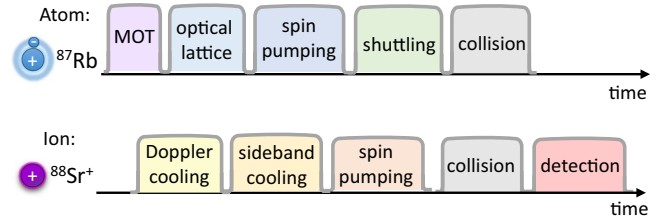

**Fig. 6 | Experimental protocol.** Neutral $^{87}$Rb atoms are prepared in the atom chamber, while short crystals of $^{87}$Rb$^+$ and $^{88}$Sr$^+$ ions are confined in the ion chamber and cooled near their motional ground state. The atomic cloud is loaded into an off-resonant optical lattice dipole trap and following state preparation is shuttled through the ion's Paul trap to enable controlled collisions.

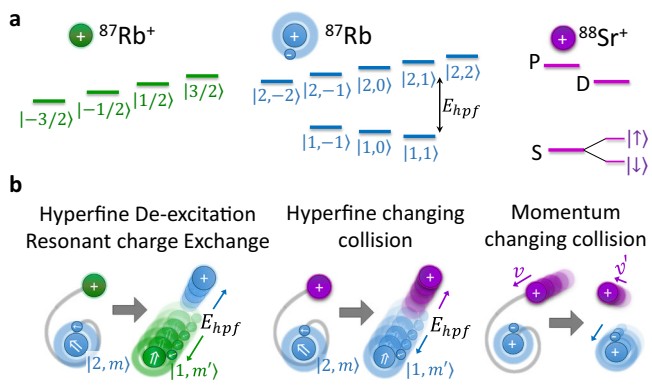

**Fig. 7 | Atom-ion cold collision processes and spin structure. a** Spin level structures of the relevant species. Left: nuclear spin states of $^{87}$Rb$^+$; middle: ground-state hyperfine manifold of $^{87}$Rb; right: electronic ground and first excited states of the spin-1/2 ion $^{88}$Sr$^+$. Prior to collisions, $^{88}$Sr$^+$ is initialized in its electronic ground state. Both $^{88}$Sr$^+$ and $^{87}$Rb$^+$ exhibit no hyperfine structure due to their zero nuclear or electron spin, respectively. **b** Collision processes relevant to this work. Left: resonant charge-exchange reactions between $^{87}$Rb and $^{87}$Rb$^+$ can de-excite the atom's hyperfine state and convert internal energy $E_{hpf}$ into kinetic energy; this process is denoted HDRCE. Middle: spin-changing collisions between $^{87}$Rb and $^{88}$Sr$^+$, primarily mediated by spin-exchange interactions, can similarly drive hyperfine de-excitation of the atom. Right: elastic collisions conserve total momentum but redistribute it between the particles. At a magnetic field of 3 G, Zeeman energy changes are negligible compared to hyperfine energy.

investigations in systems lacking large exothermic channels such as hyperfine splittings.

The techniques developed in this work, including in-situ Langevin rate calibration via momentum-changing collisions, could open opportunities for precision measurements in systems where ab initio predictions remain out of reach, such as heavy atom-ion systems like Cs$^+$-Cs. Atom-ion systems involving closed-shell alkali-metal ions with nuclear spin offer a platform for preparing and detecting well-isolated nuclear spin states, with potential applications in quantum metrology and quantum information[69–72]. In particular, systems where reactions are enhanced may enable new approaches for controlling and probing nuclear spin.

## Method
### Additional experimental details
The experiment is conducted in a dual-chamber vacuum system comprising an atom preparation chamber and an ion-trapping chamber, separated by 25 cm distance. A cloud of $^{87}$Rb atoms is first laser-cooled in a magneto-optical trap and then loaded into a far-detuned optical lattice formed by two counter-propagating laser beams.

Optical pumping and microwave pulses are used to prepare the neutral atoms in specific $|F, M\rangle$ ground-state sublevels prior to optical transport into the ion chamber. The experimental sequence is illustrated in Fig. 6. The atomic cloud is typically cooled to a few $\mu$K and is transported into the ion chamber at 24 cm/s by varying the relative optical frequencies of the lattice beams.

The ion crystal, confined in a linear Paul trap, consists of a $^{87}$Rb$^+$ ion and a $^{88}$Sr$^+$ ion, loaded via isotope-selective photoionization of neutral Sr vapor and $^{87}$Rb gas. The $^{88}$Sr$^+$ ion serves multiple roles: it provides Doppler and sympathetic cooling of all motional modes near the ground state, enables micromotion compensation, facilitates quantum logic detection of exothermic processes in $^{87}$Rb$^+$, and serves as a probe for Langevin collision calibration with neutral Rb atoms. The identity of the ion pair is routinely verified using mass spectrometry based on their secular motion frequencies.

A magnetic field of 3 G is applied during all measurements, large enough to enable spin-resolved state preparation and readout, but small compared to the hyperfine splitting (see Fig. 7a). The nuclear spin state of the $^{87}$Rb$^+$ ion is assumed to be randomized, since the neutral Rb atoms, which may reset the ion's spin via resonant charge exchange, are prepared in varying spin states over repeated experimental runs. Further details on the apparatus and experimental validation procedures are described in refs. 55,56,73.

## Logic detection of hyperfine de-excitation
We measure the probability of energetic collisions during a single passage of the neutral atom cloud using carrier-shelving thermometry based on the quantum-logic technique developed in ref. 56. This technique maps high (low) motional excitation of the ion crystal to a bright (dark) state of the logic ion, which is read out via state-dependent fluorescence. Using the same experimental parameters as in ref. 56, the protocol achieves high sensitivity ($\eta \approx 80\%$) to hyperfine energy release $E_{hpf} \sim 328 \, k_B \times$ mK, while remaining insensitive to low-energy processes ($\lesssim 1 \, k_B \times$ mK), such as trap-induced heating[27,28] or spin flips without hyperfine transitions[57].

In Fig. 2, the gray bars show the probability $P_b$ that the logic ion in a $^{88}$Sr$^+$ – $^{87}$Rb$^+$ crystal appears bright after one passage of the $^{87}$Rb cloud, indicating that the crystal is hot. To separate charge-exchange and spin-exchange contributions, we conduct additional measurements. First, we repeat the experiment using a crystal composed of two spin-up $^{88}$Sr$^+$ ions. The purple bars in Fig. 2 show the probability that at least one $^{88}$Sr$^+$ ion appears bright following a single cloud passage. Second, we estimate the readout error probability, which is associated with processes not involving hyperfine de-excitation of $^{87}$Rb atoms. To estimate these small errors, we repeat the experiments while preparing the neutral $^{87}$Rb atoms in $|1, 1\rangle$ or $|1, -1\rangle$ spin states. In these states, hyperfine transitions are endothermic and thus energetically suppressed. Even if they occurred, they would not contribute to the observed signal, which relies on exothermic energy release. The measured probabilities for these control states, shown in Fig. 2, are less than 2% per $^{88}$Sr$^+$ ion on average for both configurations. We therefore attribute this background to rare logic-ion readout errors (finite ion temperature and a ~ 0.5% per-logic-ion detection error), which scale with the number of logic ions measured and are likely independent of charge-exchange processes. Similar background levels were also observed in previous measurements with Sr$^+$ isotopes that are not optically addressed by our lasers[56].

From these measurements, we extract the hyperfine de-excitation probabilities $\bar{P}_{hpf}^{(Sr)}$ (for spin exchange) and $\bar{P}_{hpf}^{(Rb)}$ (for charge exchange) for a single $^{88}$Sr$^+$ or $^{87}$Rb$^+$ ion interacting with a neutral $^{87}$Rb atom in the upper hyperfine state $F = 2$. For each configuration, we define the average error probability as $\epsilon = \frac{1}{2}(P_b(1,1) + P_b(1, -1))$ where $P_b(F, M)$ denotes the measured probability of a bright event when the neutral $^{87}$Rb atom is initialized in $|F, M\rangle$. The probability $\bar{P}_{hpf}^{(Sr)}$ is determined from the data collected using the $^{88}$Sr$^+$ – $^{88}$Sr$^+$ crystal and is calculated

as

$$\bar{P}_{hpf}^{(Sr)} = \frac{1}{\eta \tilde{M} N_{Sr}} \sum_{M=-2}^{2} (P_b(2,M) - \epsilon), \qquad (4)$$

where $P_b$ and $\epsilon$ correspond to the purple bars in Fig. 2. This calculation subtracts the average error $\epsilon$ and averages probabilities over the $\tilde{M} = 5$ spin states in the upper hyperfine manifold. It then normalizes by the detection efficiency $\eta$ and divides by the number of logic ions $N_{Sr} = 2$.

To estimate the probability $\bar{P}_{hpf}^{(Rb)}$, we subtract the spin-exchange contribution from the $^{88}Sr^+ - ^{87}Rb^+$ crystal measurements:

$$\bar{P}_{hpf}^{(Rb)} = \frac{1}{5\eta} \left( \sum_{M=-2}^{2} (P_b(2,M) - \epsilon) \right) - \bar{P}_{hpf}^{(Sr)}, \qquad (5)$$

where $P_b$ and $\epsilon$ refer to the gray bars in Fig. 2. The extracted values of $\bar{P}hpf^{(Sr)}$ and $\bar{P}hpf^{(Rb)}$ are shown in Fig. 3a, highlighting the substantial suppression of HDRCE relative to spin exchange.

## Momentum changing collision measurements

We develop an in situ technique to estimate the rate of momentum-changing collisions, based on atom-ion dynamics under controlled excess micromotion. These collisions, elastic scattering events with large deflection angles[74,75], efficiently redistribute kinetic energy between the ion and the atom, as schematically illustrated in Fig. 7b. In contrast to glancing collisions, which are predominantly forward-scattering and transfer little energy[75,76], momentum-changing collisions produce nearly isotropic energy redistribution in the center-of-mass frame. Such events are relevant to buffer-gas cooling[74,77] and ion mobility[78]. Their rate coefficient closely tracks the Langevin collision rate[74,77], making them a robust proxy for estimating Langevin probabilities in our system.

The technique applies a static electric field, $\vec{E}_{DC}$, which displaces the ion from the radio-frequency (RF) null position by $\vec{x}$, thereby introducing excess micromotion energy $E_{EMM}$. In our trap, these quantities are given by

$$x_i = \frac{eE_{DC,i}}{m_{ion}\omega_i^2}, \qquad (6)$$

and

$$E_{EMM} = \frac{1}{16} \sum_i m_{ion} \Omega^2 q_i^2 x_i^2. \qquad (7)$$

Here, $\omega_i$ denotes the secular trap frequency along each axis ($i \in \{x, y, z\}$), $\Omega$ is the RF drive frequency, $e$ is the electron charge, and $m_{ion}$ is the ion's mass. The coefficients $q_i$ denote the trap parameters from the Mathieu equation, which describe the ion's inherent micromotion[79]. For our segmented blade trap, we use $0.45\,\text{MHz} \leq \omega_i \leq 1.5\,\text{MHz}$, $\Omega = 26.5\,\text{MHz}$, $q_y = -q_x \approx 0.14$ with negligible $q_z$.

We control the displacement $x_i$ by varying the voltage $V$ on a dedicated trap electrode to generate an electric field along the $(\hat{x} + \hat{y})/\sqrt{2}$ direction, as illustrated in Fig. 4b. The linear dependence of ion position on $V$ is calibrated by imaging the ion at different field strengths, enabling precise calibration of $E_{EMM}$ via Eq. (7). Notably, cooling and shelving efficiencies remain high even for large micromotion amplitudes. This is achieved by aligning the shelving (674 nm) and cooling (422 nm) laser beams orthogonal to the micromotion direction, and by ensuring that $\omega_i \ll \Omega$.

A momentum-changing collision transfers ion's micromotion energy to secular motion, resulting in detectable heating[80]. To observe this effect, we repeat the neutral-atom shuttling experiment while varying $E_{EMM}$. Neutral atoms are prepared in the spin-polarized

ground state $|1,1\rangle$ to suppress hyperfine-changing collisions and isolate elastic processes. Following the cloud passage, carrier-shelving thermometry is applied to map the ion crystal's motional energy to internal states of the logic ion. Crystals with elevated secular motion are detected as bright states with probability $P_b$ for a single cloud passage. Figure 4a presents $P_b$ as a function of $E_{EMM}$ for a single $^{88}Sr^+$ ion. The shaded region denotes the measured background probability (0.5%) in the absence of neutral atoms. Figure 3b shows analogous measurements for two-ion crystals in two configurations: $^{88}Sr^+ - ^{88}Sr^+$ (purple) and $^{88}Sr^+ - ^{87}Rb^+$ (green). In all configurations, we observe that larger micromotion amplitudes lead to a higher rate of momentum-changing collisions, consistent with enhanced energy transfer.

The data in Figs. 4a and 3b are used to extract the unitless and energy-independent parameter $\kappa_L$, which represents the mean number of momentum-changing (Langevin) collisions per atomic cloud passage. This is done by fitting the measured bright-state probability versus excess micromotion energy to simulations of the collision process in the trap, as detailed in the next section. Notably, the experimental determination of $\kappa_L$ does not require independent knowledge of the theoretical rate coefficient $k_L$ or of the neutral cloud's atomic column density.

## Simulation of collision processes in the trap

Numerical simulations are used to extract two key quantities from experimental data: the in-trap mean number of Langevin collisions $\kappa_L$ per cloud passage, inferred from the measurements in Figs. 3b and 4a, and the hyperfine de-excitation charge-exchange rate coefficient, $k_{HDRCE}$, inferred from Fig. 3a. Ion trajectories are modeled using the formalism of ref. 81, which accounts for secular motion, intrinsic micromotion due to the RF potential, Coulomb interactions within the ion crystal, and excess micromotion from applied static electric fields. The ion's initial secular energy is randomly sampled from the distribution[82]

$$f(E) = \frac{E^2}{2k_B^3 T^3} \exp\left(-\frac{E}{k_B T}\right). \qquad (8)$$

where $T$ denotes the initial temperature. Equation (8) is the classical energy distribution for a particle confined in a three-dimensional harmonic potential, appropriate for a trapped ion's three secular modes. The resulting density of states scales as $E^2$, in contrast to the $\sqrt{E}$ scaling of free-space. This form is general and applies to other ion species, with parameters such as mass and trap frequencies only rescaling the effective temperature $T$. We note that some atom-ion experiments have reported non-thermal Tsallis-type power-law distributions[82–84], which occur under continuous operation with many collisions and no re-cooling, when the trap exchanges energy between collisions. In our experiment, each run uses a single cloud passage (typically up to one collision) followed by re-cooling, so these non-thermal effects do not build up, and the exponential (thermal) form of Eq. (8) is valid

We assume that during a single passage of the cloud, each ion in the crystal may undergo Langevin collisions, with Poisson-distributed occurrence and a mean rate $\kappa_L$. Each Langevin collision is modeled as instantaneous and energy-independent, updating the instantaneous ion velocity $\mathbf{v}_i$ according to[56,80]

$$\mathbf{v}_i \rightarrow (1 - r + \alpha r \mathcal{R}(\varphi_L))(\mathbf{v}_i - \mathbf{v}_a) + \mathbf{v}_a, \qquad (9)$$

where $\mathbf{v}_a$ is the atom velocity sampled from a Maxwell-Boltzmann distribution at $10\,\mu\text{K}$, and $\alpha = 1$ for neutral Rb atoms prepared in the lower hyperfine manifold ($F = 1$). The mass ratio is $r = \mu/m_i \approx 0.5$, with reduced mass $\mu = m_i m_a/(m_i + m_a)$, and $\mathcal{R}(\varphi_L)$ is a rotation matrix in the collision plane with a scattering angle $\phi_L$ randomly drawn from a

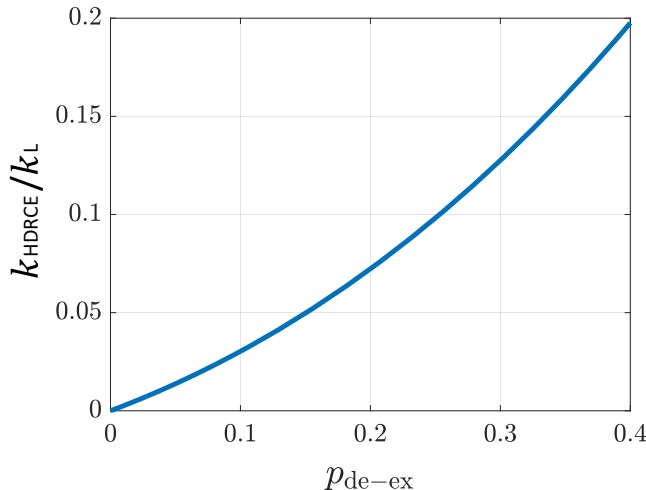

**Fig. 8 | Relationship between hyperfine de-excitation probability in the trap and in free space.** Numerical calculation of the measured hyperfine de-excitation charge-exchange probability per Langevin collision, $p_{\text{de-ex}}^{(87\text{Rb}^+)}$, in the presence of the ion trap, plotted as a function of the free-space ratio $k_{\text{HDRCE}}/k_{\text{L}}$. The enhancement of the observed charge-exchange probability in the trap arises from trap-assisted molecular bound states, which effectively increase the number of short-range encounters during a single Langevin collision[57].

classical distribution[77]. For spiraling collisions, we approximate this distribution as $f(\phi_{\text{L}}) = 0.384 - 0.013\phi_{\text{L}} - 0.014\phi_{\text{L}}^2$, for $0 \leq \phi_{\text{L}} \leq \pi$.

These collisions redistribute the ion's micromotion energy into secular motion. Denoting the secular mode amplitudes of the $^{88}$Sr$^+$ ion after a cloud passage as $A_i$ with $i \in \{x, y, z\}$, the detection probability of a bright ion is calculated as[56]

$$P_b = \cos^2\left(\frac{\pi}{2}\prod_i J_0(K_i A_i)\right), \tag{10}$$

assuming the detection pulse duration is long compared to the motional cycle. Here, $K_i$ are the components of the shelving beam's wavevector along the mode axes, and $J_0$ is the zeroth-order Bessel function. Notably, detection is sensitive only to motion along the shelving beam axis. For each parameter pair $(T, \kappa_{\text{L}})$, we simulate approximately $5 \times 10^4$ independent trajectories to obtain statistically robust averages.

The effective temperature $T$ primarily determines the bright-state probability $P_b$ at low excess micromotion energies $E_{\text{EMM}}$, while $\kappa_{\text{L}}$ primarily controls the behavior at higher $E_{\text{EMM}}$. We find good agreement with the experimental data using $T = 0.6 \pm 0.1$ mK and $\kappa_{\text{L}} = 0.29 \pm 0.02$, as shown by the black dashed curves in Figs. 3b and 4. The extracted collision temperature exceeds the energy associated with the cloud's shuttling speed ($E \approx 0.15$, $k_B \times$ mK in the center-of-mass frame), consistent with known effects of trap-induced heating[27,28]. From Fig. 3b, we also find the ratio $\kappa_{\text{L}}^{\text{Rb}^+}/\kappa_{\text{L}}^{\text{Sr}^+} = 0.97 \pm 0.06$, consistent with the theoretical expectation of unity. To test robustness, we repeated the analysis replacing Eq. (8) with the free-space Maxwell-Boltzmann distribution. While the extracted $k_L$ and $k_{\text{HDRCE}}/k_{\text{L}}$ remain unchanged, the inferred temperature shifts from 0.6 mK to 0.95 mK. These results confirm that our conclusions regarding the momentum-changing rate and charge-exchange probability are insensitive to the choice of distribution, and that the thermal state lies in the sub-mK to mK range. This uncertainty is represented by the horizontal error bar in Fig. 5 when comparing to theory.

To estimate the microscopic hyperfine de-excitation probabilities, we repeat the simulations under full excess micromotion compensation and initialize neutral $^{87}$Rb atoms in the upper hyperfine manifold ($F = 2$). In this case, the hyperfine energy release modifies the

velocity update via the scaling parameter $\alpha$ in Eq. (9). With probability $p_{\text{de-ex}}^{(X)}$, we set

$$\alpha = \sqrt{1 + 2rE_{\text{hpf}}/(m_i|\bar{v}_{\text{ion}}|^2)}, \tag{11}$$

$r(v_i - v_a)$ is the ion velocity in the center-of-mass frame, and $X \in \{^{88}\text{Sr}^+, ^{87}\text{Rb}^+\}$ denotes the colliding ion species. Fitting this model to the data in Fig. 3a, we extract $p_{\text{de-ex}}^{(87\text{Rb}^+)} = (0.053 \pm 0.04)$ per Langevin collision. This estimation agrees with the rough approximation $p_{\text{de-ex}}^{(87\text{Rb}^+)} \approx \bar{P}_{\text{hpf}}^{(\text{Rb})}/\kappa_{\text{L}}$.

To convert $p_{\text{de-ex}}^{(87\text{Rb}^+)}$ into a free-space rate coefficient, we account for trap-induced binding effects. In our trap, the loss of translational symmetry allows formation of short-lived atom-ion quasi-bound states[57,60], enabling multiple close approaches $n \geq 1$ within a single Langevin event. While these dynamics can enhance inelastic or reactive outcomes, they are not captured by the earlier simulation assuming instantaneous-collision model.

The likelihood of bound-state formation depends strongly on the initial collision energy of the atom-ion pair. For the experiments fitting $\kappa_{\text{L}}$ via momentum-changing collisions, the initial energy is sufficiently high such that quasi-bound states have negligible effect. In contrast, in the HDRCE measurement, lower collision energies favor bound-state formation. However, the release of hyperfine energy during HDRCE prevents recurrence and terminates the bound state. The free-space probability $k_{\text{HDRCE}}/k_{\text{L}}$ for HDRCE collisions relates to the observed in-trap probability through[57]

$$p_{\text{de-ex}} = \sum_n \text{PMF}(n) \sum_{m=1}^{n}\left(1 - \frac{k_{\text{HDRCE}}}{k_{\text{L}}}\right)^{m-1}\left(\frac{k_{\text{HDRCE}}}{k_{\text{L}}}\right), \tag{12}$$

where PMF($n$) is the probability mass function of $n$, the number of close encounters during the quasi-bound state lifetime. We simulate the effect of bound states in our trap using the molecular dynamics approach described in ref. 57, reconstructing PMF($n$) from approximately 2500 Langevin collisions. From this probability distribution, we derive the relationship between the microscopic exchange probability under trap conditions and the rate coefficient in free space, as shown in Fig. 8. This analysis establishes the relationship between $k_{\text{HDRCE}}$ and $k_{\text{L}}$, as discussed in the main text.

## Theoretical framework for charge-exchange collisions

We analyze hyperfine de-excitation resonant charge-exchange (HDRCE) collisions between a neutral $^{87}$Rb atom and a $^{87}$Rb$^+$ ion using a multichannel quantum scattering framework. We consider the atom in its electronic $^2S_{1/2}$ ground state and the ion in the $^1S_0$ state. The total Hamiltonian includes the adiabatic electronic interaction (treated under the Born-Oppenheimer (BO) approximation), centrifugal terms, and the atomic hyperfine interaction. In the low magnetic field regime relevant to our experiment, Zeeman splittings are small compared to the hyperfine splitting $E_{\text{hpf}}$, and the Hamiltonian is approximately isotropic. As a result, different partial waves $l$ are uncoupled. The total angular momentum $\vec{\mathcal{F}} = \vec{l} + \vec{F}_{\text{tot}}$ is conserved, and both $l$ and the total spin angular momentum $F_{\text{tot}}$ are good quantum numbers.

At large internuclear separations $R$, the system supports two energetically distinct asymptotic (fragmentation) channels, corresponding to the atom in either of its hyperfine states $F = I \pm 1/2$, and the ion in $F_{\text{ion}} = I$. These channels are separated by $E_{\text{hpf}}$ in the limit $R \to \infty$. As the atom and ion approach each other, these channels become coupled at short range due to exchange interactions, giving rise to HDRCE. The total cross section for HDRCE, for an initial atomic state $F = I + 1/2$,

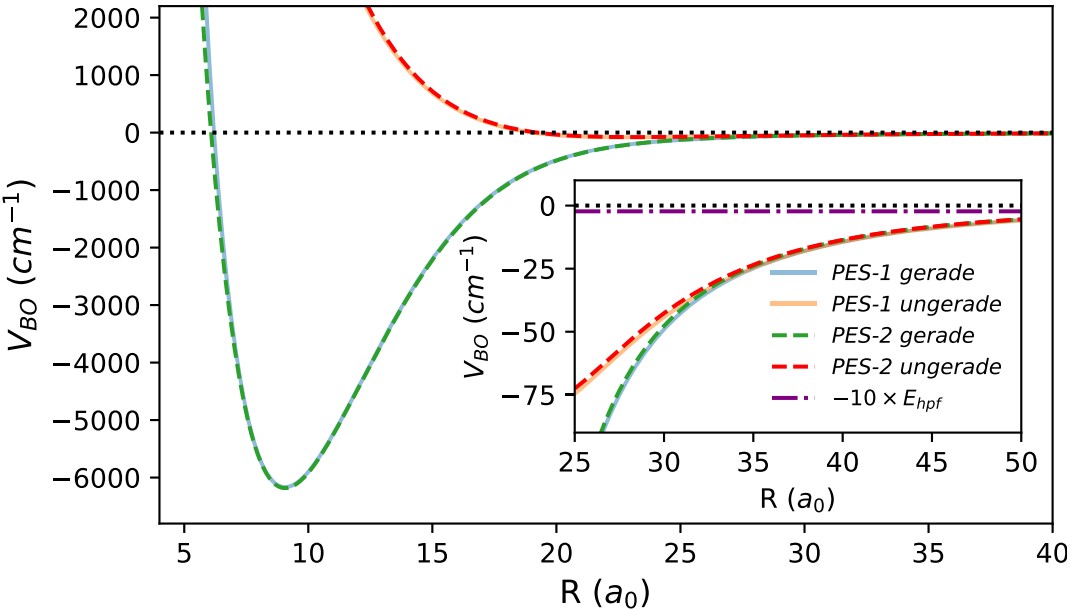

**Fig. 9 | Born-Oppenheimer potential energy curves.** Two publicly available potential energy curves (PECs) for the $^{87}$Rb $-$ $^{87}$Rb$^+$ complex are shown in the short-range regime. Solid lines denote PES-1, constructed using data accompanying ref. 86; dashed lines denote PES-2, based on the data and code provided in ref. 87. The inset highlights the region of intermediate internuclear separation where electronic wavefunction overlap decays exponentially. For reference, the atomic hyperfine energy of $^{87}$Rb, scaled by a factor of 10, is also shown.

is given by refs. 62,63

$$\sigma_{\text{HDRCE}}(E) = \frac{\pi \sum_{F_{\text{tot}}, l} (2l+1)(2F_{\text{tot}}+1)|S_{01}^{F_{\text{tot}}, l}(E)|^2}{(2I+2)(2I+1)k_i^2}, \quad (13)$$

where $S_{01}^{F_{\text{tot}}, l}(E)$ is the off-diagonal element of the scattering matrix coupling the two fragmentation channels, and $k_i = \sqrt{2\mu E_i}/\hbar$ is the wavenumber relative to the initial channel, with $E_i = E - E_{\text{hpf}}$. We use index 0(1) to represent the energetically lower (higher) fragmentation channel.

At large $R$, the BO interaction is dominated by the polarization potential, which is independent of the atom's hyperfine state[78]. As a result, the fragmentation channels are uncoupled at long range and experience identical BO potentials. Short-range coupling arises from nonzero electronic wavefunction overlap between the two nuclei, and is best described in the molecular basis, where the adiabatic potentials form a diagonal matrix with matrix elements determined by the BO potentials[62,63,85]. The transformation between fragmentation and molecular basis states is given by refs. 62,63

$$U^{F_{\text{tot}}} = \frac{(-1)^{2F_{\text{tot}}+1}}{\sqrt{2(2I+1)}}$$
$$\times \begin{pmatrix} -\sqrt{2I - F_{\text{tot}} + 1/2} & \sqrt{2I + F_{\text{tot}} + 3/2} \\ \sqrt{2I + F_{\text{tot}} + 3/2} & \sqrt{2I - F_{\text{tot}} + 1/2} \end{pmatrix}. \quad (14)$$

Figure 9 compares two publicly available BO potential curves for the $^{87}$Rb $-$ $^{87}$Rb$^+$ complex, based on independent ab initio electronic structure calculations[86,87]. While both curves agree across the full range, the inset highlights the range where the difference between the BO potentials decays exponentially with $R$ reflecting the diminishing overlap of the electronic wavefunctions. In this intermediate region, the hyperfine interaction is already at least an order of magnitude smaller than the electronic potentials.

For each partial wave $l$, the diagonal matrix elements of the Hamiltonian include a centrifugal barrier term $l(l+1)\hbar^2/2\mu R^2$, and an attractive polarization potential with $C_4 = 159.9$ a.u.[88,89]. We define $L(E)$

as the largest $l$ for which the centrifugal barrier lies below energy $E$. At a temperature of 0.6 mK, we estimate $L \approx 12$. For $l \leq L$, the short-range region is classically accessible, and HDRCE proceeds via semi-classical dynamics, potentially modified by quantum reflection near the top of the barrier[90]. For $l > L$, the short-range region becomes classically forbidden, and HDRCE proceeds through tunneling. In both regimes, resonant structures may appear in the cross section, but their prominence diminishes as the energy rises above the centrifugal barrier.

## Suppression of non-exchange mechanisms in $^{87}$Rb$^+$ $-$ $^{87}$Rb collisions

Hyperfine de-excitation in $^{87}$Rb$^+$ $-$ $^{87}$Rb collisions is dominated by resonant charge exchange, in which the valence electron is transferred between the atom and the ion, effectively swapping their identities. The relevant short-range exchange interaction is extremely strong, with a gerade-ungerade splitting that reaches hundreds to thousands of cm$^{-1}$ at the smallest separations (see Fig. 9 and ref. 87). In the absence of quantum interference effects such as s-wave suppression or partial-wave phase locking, this strong coupling would lead to a charge-exchange probability approaching 0.5 per Langevin encounter[26]. Thus, one naturally expects resonant charge exchange to dominate all other possible hyperfine-changing pathways. We note that the spin change in HDRCE transition involves the combined action of hyperfine coupling and charge exchange, and our analysis in the main text fully accounts for the hyperfine coupling component.

The next-largest corrections for HDRCE may come from spin-orbit (SO) interaction, which requires admixture of nonzero-$L$ electronic character into the short-range molecular wavefunction. In Rb$^+$, due to its closed shells the first excited electronic manifold lies about 16.5 eV[91] above the ground state, making such mixing very weak. Ab initio calculations that include relativistic effects (SO and Breit)[87] show that the net spin-dependent shift of the relevant S-manifold potentials is smaller than 1 cm$^{-1}$, about three orders of magnitude smaller than the exchange strength.

Spin rotation and magnetic dipole-dipole interactions are expected to be even more strongly suppressed. In the $\Sigma$ manifold, spin rotation arises only at second order, through combined spin-orbit and

rotational mixing with nearby $\Pi$ states[92]. Because spin-orbit effects are already extremely weak and the energy gap to the nearest $\Pi$ manifold is large, spin rotation acts only as a tiny perturbation on the spin orbit term and is negligible for HDRCE. The dipolar interaction is also strongly suppressed because it requires coupling of the nuclear spin of the $^{87}$Rb$^+$ ion to the electron spin of the neutral atom. The magnetic moment of the nucleus is about three orders of magnitude smaller than that of the electron, rendering this coupling negligible.

## Multichannel quantum defect theory modeling

We model the HDRCE process using multichannel quantum defect theory (MQDT), which provides a compact formalism to describe atom-ion scattering[62,63]. MQDT captures the complex short-range physics in a few energy-independent parameters, while treating the long-range part analytically. In the absence of partial-wave mixing, which is valid for the isotropic interaction considered here, the scattering matrix $S$ for the two open channels at each $E$ and $l$ can be expressed in terms of the real and symmetric short-range reaction matrix $K$:

$$S = (\mathbb{1} + iK)(\mathbb{1} - iK)^{-1}, \tag{15}$$

where $\mathbb{1}$ is the identity matrix. The reaction matrix $K$ incorporates both the short-range physics and the propagation through the long-range polarization potential. It is constructed via the standard MQDT relation[93]

$$K = -(Z_{fc}^c - Z_{gc}^c K^c)(Z_{fs}^c - Z_{gs}^c K^c)^{-1}. \tag{16}$$

where $K^c$ is the short-range reaction matrix and the $Z_{ab}^c$ are diagonal matrices whose elements correspond to the exact solutions of the single-channel Schrödinger equations for the long-range potential, including the hyperfine energy shifts

$$Z_{ab}^c(E,l) = \begin{pmatrix} \mathcal{Z}_{ab}^c(E,l) & 0 \\ 0 & \mathcal{Z}_{ab}^c(E - E_{\mathrm{hpf}}, l) \end{pmatrix}, \tag{17}$$

where the subscripts $ab \in \{fs, fc, gs, gc\}$, and the functions $\mathcal{Z}_{ab}^c(E,l)$ are defined in the fragmentation channel basis as given in ref. 94.

The short-range reaction matrix $K^c$ encodes all couplings that occur at distances where the electronic wavefunctions of the atom and ion overlap. To reduce the number of free parameters, we introduce two key approximations. First, we neglect hyperfine interactions at short range, justified by the fact that $E_{\mathrm{hpf}}$ is more than an order of magnitude smaller than the BO potential energy in this region (see Fig. 9, inset). In this limit, the short-range Hamiltonian is diagonal in the molecular basis. The corresponding matrix $K^{c(\mathrm{mol})}$ is then diagonal, with eigenvalues

$$K_{\pm} = \frac{1}{2}(K_g^c + K_u^c \pm (-1)^{F_{\mathrm{tot}} + l - 1/2}(K_g^c - K_u^c)). \tag{18}$$

Here, $K_g^c$ and $K_u^c$ are the single-channel reaction matrix elements for the gerade and ungerade BO potentials, respectively. These elements are related to the short-range phase shifts $\delta_{g,u}^s$ and quantum defects $\mu_{g,u}$ by[90]

$$K_{g,u}^c = \tan \delta_{g,u}^s = \tan(\pi \mu_{g,u} + \pi/4). \tag{19}$$

The reaction matrix in the fragmentation basis is obtained via the unitary transformation $K^c = U^{F_{\mathrm{tot}}\dagger} K^{c(\mathrm{mol})} U^{F_{\mathrm{tot}}}$.

Second, we assume that the short-range phases $\delta_{g,u}^s$ are approximately energy-independent across the narrow energy range relevant to cold collisions. The remaining weak dependence on the partial-wave is reflected through the change in the interaction potential which is proportional to $l(l+1)$. Expanding $\delta_{g,u}^s(E,l)$ to the first

order of $l(l+1)$, we have[62]

$$\delta_{g,u}^s(E,l) = \pi \left[ \mu_{g,u}^{(0)} + l(l+1)\beta_{g,u} + 1/4 \right], \tag{20}$$

where $\mu_{g,u}^{(0)}$ are the zero-energy, s-wave quantum defects and $\beta_{g,u}$ describe the partial-wave dependence. This approximation is well justified in the cold regime, where both the collision energy and centrifugal potential remain small compared to the short-range interaction. If higher precision is needed, for example at elevated temperatures, additional perturbative terms in energy or angular momentum can be included. Together, Eqs. (13)-(20) define the full MQDT framework used in this work to model HDRCE.

## Determination of short-range parameters and thermal averaging

The MQDT model used in this work requires four short-range parameters: the zero-energy, s-wave quantum defects $\mu_g^{(0)}$ and $\mu_u^{(0)}$, and the partial-wave dispersion coefficients $\beta_g$ and $\beta_u$. These parameters are estimated using a combination of ab initio potential energy curves and semi-classical modeling. While $\beta_g$ and $\beta_u$ can be reliably extracted from theoretical potentials, the values of $\mu_g^{(0)}$ and $\mu_u^{(0)}$, which determine the scattering lengths, require experimental input.

To estimate these parameters, we compute the short-range phase shifts $\delta_{g,u}^s$ using a WKB approximation valid in the classically allowed short-range region. The WKB phase for a given energy $E$ and partial wave $l$ is given by

$$\delta_{g,u}^{s(\mathrm{WKB})} = \int_{R_{\mathrm{min}}}^{\infty} \sqrt{2\mu(E - V_{g,u}) - \frac{l(l+1)}{R^2}} \, dR$$
$$- \int_{0^+}^{\infty} \sqrt{2\mu\left(E - \frac{C_4}{R^4}\right) - \frac{l(l+1)}{R^2}} \, dR, \tag{21}$$

where $V_{g,u}(R)$ are the gerade and ungerade BO potentials. By computing $\delta_{g,u}^{s(\mathrm{WKB})}$ for various $l$ and $E$, we fit the results to Eq. (20) to extract $\beta_{g,u}$ and $\mu_{g,u}^{(0)}$. For PES-1[86], we find $\beta_g = 2.9 \times 10^{-4}$ and $\beta_u = 4.7 \times 10^{-4}$ (with $\mu_g^{(0)} = 286.89$, $\mu_u^{(0)} = 77.38$). For PES-2[87], we obtain $\beta_g = 2.5 \times 10^{-4}$ and $\beta_u = 4.2 \times 10^{-4}$ (with $\mu_g^{(0)} = 288.12$, $\mu_u^{(0)} = 75.49$).

The parameters $\beta_g$, $\beta_u$ are perturbative and small in magnitude, making them more reliably determined from theory than the absolute values of $\mu_g^{(0)}$ and $\mu_u^{(0)}$. In particular, the difference $\beta_u - \beta_g$ is sufficiently small that it has negligible impact on the thermally averaged rate coefficient. Accordingly, we adopt the $\beta$ values from PES-2 for all calculations and verify that using PES-1 yields consistent predictions.

In contrast, theoretical estimates of $\mu_g^{(0)}$ and $\mu_u^{(0)}$ are less reliable. Due to the periodicity of the tangent function in Eq. (19), only the fractional part of each quantum defect, which requires high accuracy, practically affects scattering properties. Moreover, non-adiabatic and beyond-BO effects in the short-range region can shift the scattering lengths in ways not captured by adiabatic BO potentials. These limitations are well studied in simpler systems such as hydrogen-proton collisions[95]. As a result, precise determination of $\mu_{g,u}^{(0)}$ requires experimental input.

To compare theory with experiment, we compute the thermally averaged HDRCE rate coefficient by integrating the cross section over a Maxwell-Boltzmann distribution

$$k_{\mathrm{HDRCE}}(T) = \sqrt{\frac{8}{\pi\mu(k_{\mathrm{B}}T)^3}} \int_0^{\infty} E e^{-E/k_{\mathrm{B}}T} \sigma_{\mathrm{HDRCE}}(E) dE. \tag{22}$$

For numerical evaluation, the upper limit of the integral is truncated at $E = 10 \, k_{\mathrm{B}}T$. We estimate that the contribution of the tail beyond this cutoff is less than 0.1% of the total integral for a constant or decreasing

$\sigma_{\mathrm{HDRCE}}(E)$, making it negligible compared with model variance and experimental uncertainties.

To gain analytic intuition, we consider a semi-classical approximation[90], in which the long-range MQDT functions are approximated by $\mathcal{Z}_{fs}^c = \mathcal{Z}_{gc}^c = \cos \bar{\delta}_l^c$ and $\mathcal{Z}_{gs}^c = -\mathcal{Z}_{fc}^c = \sin \bar{\delta}_l^c$, where $\bar{\delta}_l^c$ is the semiclassical phase determined by the polarization potential. Assuming only classically allowed partial waves contribute, and neglecting quantum reflection and tunneling, the HDRCE cross section simplifies to

$$\bar{\sigma}_{\mathrm{HDRCE}}(E) = \frac{\pi}{k_i^2} \frac{l}{2l+1} \sum_{l=0}^{L(E)} (2l+1) \sin^2(\delta_g^s - \delta_u^s). \quad (23)$$

This result resembles that of ref. 65, with a key distinction: the phase difference involves the short-range phases $\delta_g^s$ and $\delta_u^s$, rather than the long-range elastic scattering phases of the fragmentation channels. Because these short-range phases are largely insensitive to energy and partial wave (see Eq. (20)), the semi-classical cross section exhibits phase-locking behavior.

Neglecting the $l$-dependence of $\delta_g^s$ and $\delta_u^s$ in Eq. (23) leads to the simplified expression in Eq. (2) describing ultracold behavior. In the opposite, high-temperature limit, where many partial waves contribute, the phase difference averages to $\sin^2(\delta_g^s - \delta_u^s) \approx 1/2$, a simplification known as the degenerate internal state approximation (DISA)[64,78]. This yields the classical limit in Eq. (3).

In Fig. 5b, we extend the model to higher temperatures, where the approximation in Eq. (20) requires more terms. To improve accuracy, we extract the energy and partial-wave dependence of the short-range phase from PES-2 and correct the low-energy limit using Eq. (21), yielding

$$\delta_{g,u}^s(E, l) \approx \delta_{g,u}^{s(\mathrm{WKB})}(E, l) - \delta_{g,u}^{s(\mathrm{WKB})}(0, 0) + \pi \mu_{g,u}^{(0)}. \quad (24)$$

## Data availability
The data displayed in Figures 2, 6, 7 are publicly accessible at this repository[96].

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

## Acknowledgements

This work was supported by the Israeli Science Foundation and the Goldring Family Foundation. We thank Marko Cetina, Maks Walewski, Mathew Frye, and Michał Tomza for fruitful discussions. O.K. acknowledges support from the U.S. Department of Energy, Office of Science, National Quantum Information Science Research Centers, Quantum Systems Accelerator.

## Author contributions

O.K., M.P., and N.A. contributed to the development of the experimental setup. O.K. performed the experiments. M.L. carried out the numerical simulations and theoretical analysis. O.K. analyzed the experimental data. R.O. supervised the research and secured funding. All authors contributed to the conception, design, interpretation, and writing of the manuscript.

## Competing interests

The authors declare no competing interests.
