## [Transparent Peer Review file · Nature Communications]

Quantum suppression of cold reactions far from the s-wave energy limit

Corresponding Author: Professor Or Katz

Version 0:

Reviewer comments:

Reviewer #1

(Remarks to the Author)

This paper presents an exciting and novel technique for measuring the in-situ Langevin rate of ion-atom collisions, which will surely help many researchers in the field. From a physics perspective, the paper demonstrates that charge exchange reactions for ion-atom collisions occur at a rate slower than predicted by the Langevin theory. The authors interpret this as the result of a destructive interference between two different reaction pathways in the phase-locking regime. However, the experimental results show a single temperature-averaged reaction rate data point, making it very difficult to determine the nature of the observations. Nevertheless, the topic is interesting and can have a significant impact on the cold chemistry community. Before acceptance, the authors should consider the following points:

1. In the introduction, the authors imply that only in the s-wave regime are Feshbach and shape resonances observed. However, that is not accurate. For instance, in crossed beam experiments, it is possible to see Feshbach resonances (<https://www.science.org/doi/abs/10.1126/science.adf9888>), shape resonances (<https://doi.org/10.1103/PhysRevResearch.4.043042>), and even novel glory undulation effects (<https://www.nature.com/articles/s41557-022-00907-2>). Even the observation of ion-atom Feshbach resonances does not happen in the pure s-wave regime. So, the authors should comment on this.
2. One reads “Our results constitute the first complete demonstration of partial-wave phase locking in a chemical reaction and establish a broadly applicable framework for exploring quantum effects in reactions beyond the reach of classical theory.” This is a powerful statement. In general, it is advisable not to overstate a finding, as it may have been indirectly observed in other experiments.
3. Why do the authors not extract the energy-dependent rate and compare it to the theoretical data?
4. For the thermal distribution, the authors use Eq. (S5), resembling a Tsallis distribution. Could the authors comment on it? In particular, how general is this specific expression, and how would it change if instead of looking into the $\text{Rb}^+ + \text{Rb}$ reaction, one looks into the $\text{Cs}^+ + \text{Cs}$ reactions?
5. Eq. (S19) for the thermal average reaction rate does not have the correct units. In the same equation, the authors used a cutoff at 10kBT . What is the convergence of the integral at that cutoff? In other words, how good are the results of the thermal average by putting an upper bound at 10kBT ?
6. The theoretical calculations show a flat temperature-dependent rate, as expected in the phase-locking regime. I agree that if your rate is below the Langevin rate, it would be due to a quantum behavior. However, with only a single data point, it is challenging to make that claim; it would be more accurately a hypothesis that requires further investigation. In addition, due to the large error bars on the experimentally determined rate, it is hard to put constraints on the short-range quantum defects too.

Minor points:

Please explain what is C4, the first time it appears.

Reviewer #2

(Remarks to the Author)

Review of the manuscript 622358 submitted to Nat.Comm. entitled "Quantum suppression of cold reactions far from the s-wave energy limit" by Or Katz, Meirav Pinkas, Nitzan Akerman, Ming Li, and Roe Ozeri.

In the manuscript, the authors explore one of the simplest possible reaction, namely resonant charge exchange between an ion and its parent atom at low temperatures, and find that the reaction rate is consistent with quantum interference suppressing charge exchange even at energies/temperatures high above the s-wave regime. Their measurements are obtained using a quantum-logic detection scheme on a single atom-ion pair using 88Sr^+ , since direct imaging of 87Rb^+ cannot be performed due to the closed electronic shell preventing cycling-transition. They find that the resonant charge exchange rate between 87Rb and 87Rb^+ is over an order of magnitude smaller relative to the expected rate; since their measurements take place at temperatures in the mK range, 3 orders of magnitude above the s-wave regime, and that several partial waves (roughly 12) contribute, one would expect the rate to agree with the classical value. They trace this suppressed behavior to quantum interference arising from phase locking of partial waves.

Overall, the manuscript is well written and presents results that are not only relevant to ultracold atom-ion scattering, but also to reactions where quantum interference may play an important role. It uses a quantum-logic detection scheme that is very sensitive to state of the system, and that can be used precisely at fairly low scattering energy, opening the door to measure charge exchange in system that cannot be optically probed. I find their analysis convincing (especially with the details given in the Method section) and recommend publication once the authors have addressed a few points below.

First, some statements come across a bit too strong, like in the abstract "Here, we report the first observation of quantum interference suppressing a chemical reaction ...": there are a few recent papers, like by Kang-Kuen Ni's group (see list at the end). So, maybe stating that it the first (to my knowledge, but should be checked !) involving a charge exchange reaction, or simply removing the word "first".

Also in the abstract and elsewhere in the text (e.g., 1st sentence in "Experimental Study" on p.4), the authors use "... 87Rb atom and its parent ion 87Rb^+ ...": the standard way (at least if referring to older 1958 paper by Dalgarno cited as [54] in the manuscript) would be "... 87Rb^+ ion in its parent atom 87Rb ..."

The authors should be careful with general statements, like in the introduction (right column, middle of the page: "In contrast, inelastic collisions and chemical reactions are predominantly governed by short-range molecular dynamics, where atomic wavefunctions overlap substantially and relative scattering phase shifts across relevant channels vary more uniformly with partial waves.") this is not always true, in the sense that coupling needs to be significant enough, and wave function do not always overlap substantially (e.g., the inner walls and potential wells do not overlap substantially as shown in Fig.1). The relative phase shifts might vary more uniformly for a given range of energy and/or partial waves, but can become quite different outside specific range. I would suggest to change this statement to something like "In contrast, inelastic collisions and chemical reactions are predominantly governed by short-range molecular dynamics, where coupling between atomic wavefunctions becomes substantial ..."

Could the authors be more explicit about how you got the Langevin rate $k_L = 2.45 \times 10^{-9} \text{ cm}^3/\text{s}$? A rate is usually obtained by thermal averaging a cross section (I assume here the Langevin ... please give its definition) and a velocity distribution. I am not clear on it (I see velocity distributions in the Method, but not clear if it relates to k_L). What was the temperature? Maybe I am not clear about the discussion of Fig.4a in Method. Also, the exact definition of the Langevin cross section may differ by a factor of 2 depending on the authors, and such a factor might reduce the "suppression" by such a factor in a similar fashion, could the authors be more explicit about equation (S17)? It seems to be semi-classical in origin (at least the $(l+1/2)$ and the $1/4$ terms point to it).

When discussing lighter species and Yb studies in "Discussion and Outlook" on p.7, the authors should also cite the work of Dalgarno's group [see PRA 80, 030703 (2009) and PCCP 13, 19026 (2011) listed below]. I also list a few additional references, like the review paper by Cote (could be cited with [18] for example).

Finally, there are a few points I would like the authors to consider that would strengthen the paper (although it might require additional work). First, could the authors repeat their measurements with 85Rb instead? Since the phase shifts $\mu(0)g/u$ would be different, the suppression (or lack) would be different (i.e. the quantum interference should lead to different observation). Furthermore, since the electronic potential curves are basically identical for both isotopes, by adjusting the potential curves to match the $\mu(0)g/u$ of 87Rb , they could determine/calculate the values for 85Rb , and hence predict the impact of quantum interference (and then confirm/measure it).

Another interesting question is what happens when the magnetic field is increased. Right now, it seems that the Zeeman energy is negligible. What would happen if it was not the case? Would some of the hyperfine channels close up for the exchange? Could the interference be controlled by changing the B-field? It would be interesting for the authors to comment/consider this possibility.

So, again, this is an excellent manuscript, and I recommend its publication once the authors have considered the comments above and added the references below.

List of missing references to be added in chronological order:

- P. Zhang, A. Dalgarno, and R Côté, "Scattering of Yb and Yb+", PRA 80, 030703(R) (2009).
- P. Zhang, A. Dalgarno, R. Côté, and E. Bodo "Charge exchange in collisions of beryllium with its ion", Phys. Chem. Chem. Phys. 13, 19026 (2011).
- R Côté "Chapter Two - Ultracold Hybrid Atom-Ion Systems", Editor(s): Ennio Arimondo, Chun C. Lin, Susanne F. Yelin, in Advances In Atomic, Molecular, and Optical Physics, Academic Press, Vol.65, pp. 67-126 (2016).
- Y.-X. Liu, L.Zhu, J. Luke, J.J.A. Houwman, M.C. Babin, M.-G. Hu, and K.-K. Ni "Quantum interference in atom-exchange reactions", Science 384, 1117-1121 (2024).
- H. da Silva, B. K. Kendrick, H. Li, S. Kotochigova, and N. Balakrishnan, "Nonadiabatically Driven Quantum Interference Effects in the Ultracold $K + KRb \rightarrow Rb + K$ 2 Chemical Reaction", J. Phys. Chem. Lett., 16 6171 (2025).
- S. Haze, J.-L. Li, D. Dorer, J.P. D'Incao, P.S. Julienne, E. Tiemann, M. Deiss, and J. Hecker Denschlag, "Controlling few-body reaction pathways using a Feshbach resonance", Nature Phys, 21, 228 (2025).
- I.Simbotin and R. Côté "Quantum correction to the Langevin cross section in resonant-exchange processes" arXiv:2508.09302 (2025).

Reviewer #3

(Remarks to the Author)

Referee report on „Quantum suppression of cold reactions far from the s-wave energy limit“ by Or Katz et al.

In this nice work, the authors investigate inelastic collisions between an ultracold Rb atom (in the electronic ground state) with a Rb⁺ ion at a cold, but finite temperature. When the Rb atom is initially in a state with total angular momentum $F = 2$, it can relax to the state $F = 1$ in the inelastic collision. The authors find that the relaxation rate for the collisions with Rb⁺ is much lower than expected from simple statistical/ semi-classical arguments for the given temperature. At finite temperatures the atom ion reactive collision can take place within a range of collisional angular momenta. The reaction cross sections for the various angular momenta would normally be expected to vary randomly within a given range. This would result in a total cross section which is the product of the number of contributing partial waves times the average cross section within the range. For the specific reaction considered here, however, this averaging out does not occur, since the cross sections for the partial waves are nearly independent of the collisional angular momentum. This phenomenon is called phase locking. Phase locking has been predicted in recent years and has been subsequently experimentally observed to various degrees. The observation, discussed in the present work, is particularly clear. The authors claim that their results "constitute the first complete demonstration of partial-wave phase locking in a chemical reaction".

In order to show that the relaxation rate in the Rb / Rb⁺ collision is much smaller than the expected average, the authors compare the observed cross sections to spin changing collisions of Rb with a Sr⁺ ion where phase locking does not occur and averaging takes place.

The quality of the presented experimental and theoretical work is very high. The paper is well written. I think that it is generally worthy of being published in Nature Communications.

I have a number of questions and comments which should be resolved before the work can be accepted:

- a) The authors state that the hyperfine deexcitation in the Rb-Rb⁺ collision happens in the course of charge exchange between the two atoms; i.e. where the valence electron hops between the two atoms. Besides the spin-exchange interaction, the Rb spin could in principle also flip due to spin- dipolar interaction. Can we exclude such other processes? For the relaxation in the collision with Sr⁺ the authors state that there can be variety of physical processes that can contribute to this effect.
- b) The authors stress how important the calibration relative to the Langevin collision is. Why is it not enough to simply show (as done in Fig. 2a) that the inelastic rate for the case of Rb⁺ is much smaller than for Sr⁺? Both rates are limited by the Langevin cross section, because the processes occur at close range. If one is much smaller than the other one, then this should be due to phase locking, right?
- c) The calibration of the Langevin cross section involves several semiclassical models where e.g. the trajectories of the ion through the gas of atoms are simulated. For this, an accurate treatment of all kinds of excess micromotion, of the trap induced bound state effects, and other physics is needed. The system is really quite complex. Given the systematic uncertainties in all of these models, I wonder to what extent all of this leads to a sizeable uncertainty of the calibration.
- d) In Fig. 2b) the probability P_b increases with the micromotion. Is this increase mainly explained by the fact that in a collision with higher energy collision there is a higher probability to impart a large enough energy so that the optical shelving transition does not work anymore, or is there also an additional effective increase in the collision cross section?
- e) According to equation (3) the rate constant converges towards $l/(2 * l + 1) k_L$. If I use $l = 3/2$, this results in $3/8 k_L$, not $3 / 16 k_L$, as stated in the text. What am I missing here?
- f) In Figure 3a) the experimental measurements are represented by a single dot at a particular temperature. How come in Figure 3b) these experimental measurements are then represented by a black line over a large range temperature range?
- g) In the caption of Fig. 1 it reads "the contributions from different partial waves sum coherently and quantum interference can persist beyond the s-wave limit". I am not happy how this sentence is formulated, as it is misleading. In quantum mechanics amplitudes might sum coherently, but not rates or cross sections. Quantum interference persists of course beyond the s-wave limit, but the effects of quantum interference might wash out.
- h) In the "extended data Fig. 3", the measured loss probabilities for Rb in the state $F = 1$ are still significantly lower for Rb⁺ than for Sr⁺. How can this be explained? From my understanding, the charge exchange suppression should not lead to any effects here.

i) In the discussion the authors write: "Trap-induced dynamics, while capable of enhancing reaction rates [48, 51], cannot account for the observed suppression and are explicitly included in our analysis." This statement might be correct in the classical description, but would it also generally be correct in a quantum description where interference effects might take place?

j) In the outlook the authors write: "The techniques developed here, including in-situ Langevin rate calibration via momentum-changing collisions, open new opportunities for precision measurements in systems beyond the reach of current theoretical methods." Can they explain more what they mean?

k) Reference [49] of the authors is not up-to-date, as it has been already published in Science Advances.

l) In the references there are repeated problems with missing capital letters.

Version 1:

Reviewer comments:

Reviewer #1

(Remarks to the Author)

The authors have address all my comments so it should be published.

Reviewer #2

(Remarks to the Author)

Review of the resubmitted manuscript:

After reading carefully the comments of all reviewers and the corresponding responses, I am satisfied with the changes made. In particular, the authors have answered all my questions and followed the suggestions made. As a result, I find the manuscript improved and ready for publication.

I recommend publication as is.

Reviewer #3

(Remarks to the Author)

The authors have done a very good job in answering/ fixing the issues raised by all three referees. I have no more concerns but recommend the manuscript to be published as is.

made.

Reviewer #1 (Remarks to the Author):

This paper presents an exciting and novel technique for measuring the in-situ Langevin rate of ion-atom collisions, which will surely help many researchers in the field. From a physics perspective, the paper demonstrates that charge exchange reactions for ion-atom collisions occur at a rate slower than predicted by the Langevin theory. The authors interpret this as the result of a destructive interference between two different reaction pathways in the phase-locking regime. However, the experimental results show a single temperature-averaged reaction rate data point, making it very difficult to determine the nature of the observations. Nevertheless, the topic is interesting and can have a significant impact on the cold chemistry community.

We thank Reviewer #1 for their thoughtful review and for recognizing the novelty and potential impact of our measurement technique. We address the points about temperature averaging and interpretation below.

Before acceptance, the authors should consider the following points:

1. In the introduction, the authors imply that only in the s-wave regime are Feshbach and shape resonances observed. However, that is not accurate. For instance, in crossed beam experiments, it is possible to see Feshbach resonances (<https://www.science.org/doi/abs/10.1126/science.adf9888>), shape resonances (<https://doi.org/10.1103/PhysRevResearch.4.043042>), and even novel glory undulation effects (<https://www.nature.com/articles/s41557-022-00907-2>). Even the observation of ion-atom Feshbach resonances does not happen in the pure s-wave regime. So, the authors should comment on this.

We thank the reviewer for this important comment. We have revised the Introduction to explicitly note that Feshbach and shape resonances, as well as related quantum features, can also occur at higher energies and higher partial waves, citing the reviewer's suggested references. The revised sentence now reads: *"In this regime, quantum phenomena such as shape [] and Feshbach resonances[] are especially pronounced and tunable; though such resonances can also occur at higher energies and higher partial waves in crossed-beam and related experiments []."*

In addition, we have added a clarifying sentence to emphasize that while such resonances may appear in individual channels, their signatures are typically averaged out in total thermally averaged reaction rates. This distinction highlights our research focus on coherent interference in a thermally averaged chemical reaction rate deep in the multi-partial-wave regime, involving comparable contribution from roughly a dozen partial waves.

2. One reads "Our results constitute the first complete demonstration of partial-wave phase locking in a chemical reaction and establish a broadly applicable framework for exploring quantum effects in reactions beyond the reach of classical theory." This is a powerful statement. In general, it is advisable not to overstate a finding, as it may have been indirectly observed in other experiments.

We agree with the referee that, given the single temperature-averaged data point and the possibility of other related observations, our original wording was too strong. We have revised the sentence to more cautiously state: *"Our results provide signatures of partial-wave phase locking in a chemical reaction..."*.

3. Why do the authors not extract the energy-dependent rate and compare it to the theoretical data?

We thank the reviewer for this question. In our setup each measurement is thermally averaged over a narrow but finite collision-energy distribution set by cloud velocity, ion initial energy and trap dynamics, so we extract a single point at the mean temperature. Mapping out the energy-dependent rate would require systematically varying the collision energy while maintaining high statistics and calibration accuracy, which is experimentally very challenging. Increasing the atom cloud velocity shortens the interaction time, lowers the Langevin rate and thus the signal without reducing background, while at the same time our logic-spectroscopy readout becomes less reliable at higher energies due to increased momentum-changing scattering, which alters the shelving probability regardless of whether a charge exchange reaction occurs. These factors make energy-resolved measurements prone to large uncertainties with our present setup and protocol, particularly for rare events such as charge exchange. Future improvements, such as optical trapping or techniques to momentarily switch off trap fields during collisions, could enable more accurate energy-resolved studies in future work.

Following this comment, we have added the following sentences to the Outlook section: *“Looking ahead, improved trapping and detection methods could enable energy-resolved measurements of charge-exchange reaction rates across a broader range of collision energies and temperatures. Approaches such as optical trapping or temporarily switching off trap fields during collisions would lower the minimum achievable temperature and complement the thermally averaged results presented here.”*

4. For the thermal distribution, the authors use Eq. (S5), resembling a Tsallis distribution. Could the authors comment on it? In particular, how general is this specific expression, and how would it change if instead of looking into the $\text{Rb}^+ + \text{Rb}$ reaction, one looks into the $\text{Cs}^+ + \text{Cs}$ reactions?

We thank the reviewer for this thoughtful question and agree that additional clarification of the thermal distribution is helpful. Equation (S5) represents the classical energy distribution for a particle in a three-dimensional harmonic potential, which is appropriate because the ion is confined in a trap. In this case, the ion’s secular motion consists of three independent harmonic modes, leading to a density of states that scales as E^2 rather than the \sqrt{E} scaling that appears for particles with a Maxwell-Boltzmann distribution in free space. As a result, the probability distribution for the total secular energy naturally takes the form of Eq. (S5). This functional form is general and applies to other ion species (e.g., Cs^+), with species-dependent parameters such as the ion mass affecting the effective temperature but not altering the overall structure of the distribution.

Tsallis statistics, in contrast, describe non-thermal states that can emerge in Paul traps after many collisions without re-cooling, due to energy injected or removed by the trap between collisions. As shown for example in Ref. [PRL 118 143401 (2017)], repeated collisions with variable heating rates gradually build up a distribution with a power-law tail, which is well described by a Tsallis (q-exponential) function. In our experiment, however, each run consists of a single atomic cloud passage followed by re-cooling before the next experiment, ensuring that each collision is independent and preventing the accumulation of energy injected by the trap. In this single-collision limit, the Tsallis framework for trapped ions smoothly reduces to the exponential form of Eq. (S5), so a Tsallis description is not required. To make this clearer for the reader, we have added a brief explanation in the Methods section stating why the harmonic-oscillator form density of states is appropriate, how it differs from a free-space Maxwell-Boltzmann distribution, and under what conditions a Tsallis distribution would become relevant.

Finally, we note that our results are relatively robust to the specific choice of distribution. As a test, we repeated the analysis using the free-space Maxwell-Boltzmann form instead of Eq. (S5). We find that the

extracted ratio k_{HDRCE}/k_L is unchanged within uncertainties but that for that distribution $T=0.95\text{mK}$, compared to $T=0.6\text{mK}$ using Eq. (S5). This confirms that the effective thermal state lies in the sub-mK to mK regime and demonstrates that our conclusions are insensitive to the exact distribution assumed. This robustness is now explicitly mentioned in the Methods section.

5. Eq. (S19) for the thermal average reaction rate does not have the correct units. In the same equation, the authors used a cutoff at $10k_B T$. What is the convergence of the integral at that cutoff? In other words, how good are the results of the thermal average by putting an upper bound at $10k_B T$?

We thank the reviewer for raising this point and for spotting the units typo. We have corrected the units of Eq. (S19) in the revised text. For numerical evaluation, we estimate that truncating the integral above $10k_B T$ changes the result by less than 0.1% for constant or decreasing energy-dependent cross section, negligible relative to model variance and experimental uncertainties. We have clarified this point in the revised Methods section under Eq. (S19).

6. The theoretical calculations show a flat temperature-dependent rate, as expected in the phase-locking regime. I agree that if your rate is below the Langevin rate, it would be due to a quantum behavior. However, with only a single data point, it is challenging to make that claim; it would be more accurately a hypothesis that requires further investigation. In addition, due to the large error bars on the experimentally determined rate, it is hard to put constraints on the short-range quantum defects too.

We thank the reviewer for this thoughtful comment. We agree that with only a single temperature-averaged data point, our conclusions should be stated more cautiously. In the revised manuscript, we have softened our language to avoid overclaiming, now referring to our results as signatures of coherent quantum interference rather than a definitive observation. These changes have been made throughout the manuscript including the abstract, introduction, and discussion.

While we cannot fully determine the short-range quantum defects, the strong suppression we observe below the Langevin rate is consistent with the phase-locking regime only when the difference between the quantum defects $|\mu_g^{(0)} - \mu_u^{(0)}|$ is small compared to unity. Thus, even a single data point constrains the difference between the gerade and ungerade scattering channels. This can be seen qualitatively in Fig. 3a, where the data agree well with the green models but not with the orange and blue models, and more quantitatively Fig. 3b shows that, within our error bars, $|\mu_g^{(0)} - \mu_u^{(0)}| < 0.1$.

In the revised manuscript, we now explicitly highlight in the Discussion section that our observations do not provide full information on the s -wave scattering lengths but rather constrain only the relative difference between the quantum defects.

Minor points: Please explain what is C4, the first time it appears.

Done. We now define C4 at its first appearance.

We thank the reviewer again for their careful reading and helpful suggestions, which have improved the clarity of the manuscript.

Reviewer #2 (Remarks to the Author):

Review of the manuscript 622358 submitted to Nat.Comm. entitled “Quantum suppression of cold reactions far from the s-wave energy limit” by Or Katz, Meirav Pinkas, Nitzan Akerman, Ming Li, and Roe Ozeri.

In the manuscript, the authors explore one of the simplest possible reaction, namely resonant charge exchange between an ion and its parent atom at low temperatures, and find that the reaction rate is consistent with quantum interference suppressing charge exchange even at energies/temperatures high above the s-wave regime. Their measurements are obtained using a quantum-logic detection scheme on a single atom-ion pair using 88Sr^+ , since direct imaging of 87Rb^+ cannot be performed due to the closed electronic shell preventing cycling-transition. They find that the resonant charge exchange rate between 87Rb and 87Rb^+ is over an order of magnitude smaller relative to the expected rate; since their measurements take place at temperatures in the mK range, 3 orders of magnitude above the s-wave regime, and that several partial waves (roughly 12) contribute, one would expect the rate to agree with the classical value. They trace this suppressed behavior to quantum interference arising from phase locking of partial waves.

Overall, the manuscript is well written and presents results that are not only relevant to ultracold atom-ion scattering, but also to reactions where quantum interference may play an important role. It uses a quantum-logic detection scheme that is very sensitive to state of the system, and that can be used precisely at fairly low scattering energy, opening the door to measure charge exchange in system that cannot be optically probed. I find their analysis convincing (especially with the details given in the Method section) and recommend publication once the authors have addressed a few points below.

We thank Reviewer #2 for the positive assessment, and for finding our results broadly relevant and our methodology convincing. We appreciate the recommendation for publication and address the constructive points raised below.

First, some statements come across a bit too strong, like in the abstract “Here, we report the first observation of quantum interference suppressing a chemical reaction ...”: there are a few recent papers, like by Kang-Kuen Ni’s group (see list at the end). So, maybe stating that it the first (to my knowledge, but should be checked !) involving a charge exchange reaction, or simply removing the word “first”.

We thank the reviewer for this helpful comment. We have now added citations to all relevant works suggested by the reviewer, including recent studies by Kang-Kuen Ni’s group. While, to our understanding, those studies involve only a few partial waves, we agree that softening our wording improves clarity and avoids the possibility of overstating novelty. Accordingly, we have removed any mention of “first” and now describe our findings as “signatures” of quantum interference rather than a definitive “first” observation.

Also in the abstract and elsewhere in the text (e.g., 1st sentence in “Experimental Study” on p.4), the authors use “ ... 87Rb atom and its parent ion 87Rb^+ ...”: the standard way (at least if referring to older 1958 paper by Dalgarno cited as [54] in the manuscript) would be “ ... 87Rb^+ ion in its parent atom 87Rb ...”

We thank the reviewer for this suggestion; we have revised the text and abstract to use the standard phrasing throughout.

The authors should be careful with general statements, like in the introduction (right column, middle of the page: “In contrast, inelastic collisions and chemical reactions are predominantly governed by short-range molecular dynamics, where atomic wavefunctions overlap substantially and relative scattering phase shifts across relevant channels vary more uniformly with partial waves.”) this is not always true, in the sense that coupling needs to be significant enough, and wave function do not always overlap substantially (e.g., the inner walls and potential wells do not overlap substantially as shown in Fig.1). The relative phase shifts might vary more uniformly for a given range of energy and/or partial waves, but can become quite different outside specific range. I would suggest to change this statement to something like “In contrast, inelastic collisions and chemical reactions are predominantly governed by short-range molecular dynamics, where coupling between atomic wavefunctions becomes substantial ...”

We thank the reviewer for this suggestion and have revised the sentence as recommended.

Could the authors be more explicit about how you got the Langevin rate $k_L = 2.45 \times 10^{-9} \text{ cm}^3/\text{s}$? A rate is usually obtained by thermal averaging a cross section (I assume here the Langevin ... please give its definition) and a velocity distribution. I am not clear on it (I see velocity distributions in the Method, but not clear if it relates to k_L). What was the temperature? Maybe I am not clear about the discussion of Fig.4a in Method. Also, the exact definition of the Langevin cross section may differ by a factor of 2 depending of the authors, and such a factor might reduce the “suppression” by such !

We thank the referee for this question. In the revised Methods, we now define the classical Langevin rate coefficient for the polarization potential $V(R) = -C_4/R^4$ as $k_L = 2\pi\sqrt{C_4/\mu}$ where μ is the atom-ion reduced mass and C_4 is the atom-ion polarization coefficient set by the neutral atom’s static polarizability. Because C_4 is constant, k_L is independent of both energy and temperature, so no thermal averaging is required. For the Rb-Rb+ system this gives $k_L = 2.45 \times 10^{-9} \text{ cm}^3/\text{s}$. We now explicitly indicate our convention and value for C_4 to avoid the factor of 2 ambiguity in literature.

Extended Data Fig. 4a is not used to compute k_L . Instead, it is used to experimentally determine the in-trap quantity κ_L , which is the mean number of momentum-changing (Langevin) collisions per atomic cloud passage; this approach enables calibration of the total number of Langevin collisions without requiring independent knowledge of the atomic cloud density. We extract κ_L and the effective collision temperature T by fitting the measured bright-state probability versus excess micromotion energy with our molecular-dynamics model. The velocity distributions shown in the Methods are used only within this fitting procedure to model the trapped-ion dynamics and determine T ; they play no role in the calculation of k_L .

In our analysis, κ_L provides an in-situ calibration, allowing us to normalize the measured HDRCE probability relative to the total rate of Langevin collisions in the trap without needing external knowledge of the cloud’s column density. The MQDT theory also produces HDRCE relative to the total cross section and uses the same k_L to convert these relative values into the absolute vertical scale shown in Fig. 3. Thus, while k_L sets the absolute scale, the conclusion of strong suppression relies on the ratio of HDRCE to total Langevin collisions and remains unaffected by the precise value of k_L .

To clarify this point, in the revised manuscript we: (i) explicitly defined the energy-independent Langevin rate coefficient k_L in the main text where first presented, (ii) added a note to the caption of Fig. 3 stating that the value of k_L only affects the absolute scale and does not influence the relative comparison of the

theoretical and experimental results, and (iii) clarified in the Methods section, around Extended Data Fig. 4a, that these data are used to extract κ_L (the in-trap average collision rate) and not to compute k_L .

in a similar fashion, could the authors be more explicit about equation (S17) ? It seems to be semi-classical in origin (at least the $(l+1/2)$ and the $\frac{1}{4}$ terms point to it).

We thank the referee for this question. In our model, the partial wave l affects only the interaction potential V through the centrifugal term. Taking the $l = 0$ Hamiltonian as the reference (unperturbed) case, the additional contribution for $l > 0$ scales as $l(l + 1)$, which motivates a Taylor expansion of the short-range quantum defect in powers of this quantity. The choice between expressing this dependence as $l(l + 1)$ or $(l + \frac{1}{2})^2$ is not significant, as both lead to the same l -dependence and therefore to the same slope parameter β ; the two conventions differ only by a constant shift of $\mu(l = 0)$ by $\beta/4$. For consistency with prior work [PRA 89, 052704 (2014)], we now adopt the $l(l + 1)$ form in Eq. S17 and have added explanatory text referencing this treatment. This change is purely notational and does not affect any results, apart from a constant offset $\beta/4$ (which is numerically very small).

When discussing lighter species and Yb studies in “Discussion and Outlook” on p.7, the authors should also cite the work of Dalgarno’s group [see PRA 80, 030703 (2009) and PCCP 13, 19026 (2011) listed below]. I also list a few additional references, like the review paper by Cote (could be cited with [18] for example).

We thank the Reviewer for this helpful suggestion. We have now incorporated citations to all of the recommended references in the relevant sections of the manuscript.

Finally, there are a few points I would like the authors to consider that would strengthen the paper (although it might require additional work). First, could the authors repeat their measurements with ^{85}Rb instead ? Since the phase shifts $\mu(0)g/u$ would be different, the suppression (or lack) would be different (i.e. the quantum interference should lead to different observation). Furthermore, since the electronic potential curves are basically identical for both isotopes, by adjusting the potential curves to match the $\mu(0)g/u$ of ^{87}Rb , they could determine/calculate the values for ^{85}Rb , and hence predict the impact of quantum interference (and then confirm/measure it).

We appreciate the suggestion to repeat the measurement with ^{85}Rb . This is not feasible in our present apparatus, which uses an enriched ^{87}Rb source and would require major re-engineering, including invasive modifications of the source and vacuum system. From a theoretical perspective, while mass-scaling can serve as a useful starting point, a robust prediction for HDRCE would also require a more comprehensive treatment that includes the hyperfine structure. Since our current setup cannot probe these effects experimentally, we believe this valuable extension is best suited for a dedicated future project beyond the scope of this work (and not necessarily limited to the Rb system).

To highlight this direction, we have added the following note to the Discussion section: *“Furthermore, studying charge exchange between different isotopes, such as ^{85}Rb - $^{85}\text{Rb}^+$ or other elements, is a compelling direction that could reveal how changes in reduced mass and hyperfine structure influence phase-locking and suppression, providing a stringent test of the theoretical framework”.*

Another interesting question is what happens when the magnetic field is increased. Right now, it seems that the Zeeman energy is negligible. What would happen if it was not the case ? Would some of the

hyperfine channels close up for the exchange ? Could the interference be controlled by changing the B-field ? It would be interesting for the authors to comment/consider this possibility.

We thank the reviewer for this insightful suggestion and agree that the magnetic field is a useful control knob (indeed, we have previously used it to suppress channels via energy conservation and to study trap-induced molecules [Nat. Phys. 19, 1573 (2023)]). In the present work, the HDRCE channel is exothermic by the ^{87}Rb hyperfine splitting (with hyperfine energy splitting of $E_{\text{hpf}} \sim 6.8$ GHz). To close or substantially shift this channel would require Zeeman shifts comparable to E_{hpf} , i.e. multi-kilogauss fields, far beyond our apparatus capabilities (up to ~ 20 G). Moreover, $^{87}\text{Rb}^+$ is closed-shell ion, so its Zeeman response is set by the nuclear moment and is negligible on this scale.

That said, moderate fields (tens of gauss) can re-weight Zeeman subchannels at the MHz level. While our present thermometry is optimized for detecting large energy releases and is less sensitive to small Zeeman-scale energies, re-tuning the logic readout thermometry could make us sensitive to Zeeman-scale exothermicities, potentially enabling spin-selective logic detection of other channels, facilitated by the magnetic field.

We now detail this possibility in the Discussion section: *“Finally, applying modest magnetic fields (on the order of tens of gauss) could shift the populations of different Zeeman spin states. With appropriate adjustments to the logic detection scheme, this could open the door to spin-selective studies of resonant exchange processes, or enable investigations of systems lacking large exothermic channels such as hyperfine splittings”.*

So, again, this is an excellent manuscript, and I recommend its publication once the authors have considered the comments above and added the references below.

We thank the reviewer again for their careful reading and for the constructive comments and reference suggestions, which have helped improve the clarity and impact of the manuscript.

List of missing references to be added in chronological order:

- P. Zhang, A. Dalgarno, and R. Côté, “Scattering of Yb and Yb⁺”, PRA 80, 030703(R) (2009).
- P. Zhang, A. Dalgarno, R. Côté, and E. Bodo “Charge exchange in collisions of beryllium with its ion”, Phys. Chem. Chem. Phys. 13, 19026 (2011).
- R. Côté “Chapter Two - Ultracold Hybrid Atom-Ion Systems”, Editor(s): Ennio Arimondo, Chun C. Lin, Susanne F. Yelin, in Advances In Atomic, Molecular, and Optical Physics, Academic Press, Vol.65, pp. 67-126 (2016).
- Y.-X. Liu, L. Zhu, J. Luke, J.J.A. Houwman, M.C. Babin, M.-G. Hu, and K.-K. Ni “Quantum interference in atom-exchange reactions”, Science 384, 1117-1121 (2024).
- H. da Silva, B. K. Kendrick, H. Li, S. Kotochigova, and N. Balakrishnan, “Nonadiabatically Driven Quantum Interference Effects in the Ultracold $\text{K} + \text{KRb} \rightarrow \text{Rb} + \text{K}_2$ Chemical Reaction”, J. Phys. Chem. Lett., 16 6171 (2025).
- S. Haze, J.-L. Li, D. Dorer, J.P. D’Incao, P.S. Julienne, E. Tiemann, M. Deiss, and J. Hecker Denschlag, “Controlling few-body reaction pathways using a Feshbach resonance”, Nature Phys, 21, 228 (2025).
- I. Simbotin and R. Côté “Quantum correction to the Langevin cross section in resonant-exchange processes” arXiv:2508.09302 (2025).

All of these references have been added to the revised manuscript.

Reviewer #3 (Remarks to the Author):

Referee report on „Quantum suppression of cold reactions far from the s-wave energy limit“ by Or Katz et al.

In this nice work, the authors investigate inelastic collisions between an ultracold Rb atom (in the electronic ground state) with a Rb⁺ ion at a cold, but finite temperature. When the Rb atom is initially in a state with total angular momentum $F = 2$, it can relax to the state $F = 1$ in the inelastic collision. The authors find that the relaxation rate for the collisions with Rb⁺ is much lower than expected from simple statistical/ semi-classical arguments for the given temperature. At finite temperatures the atom ion reactive collision can take place within a range of collisional angular momenta. The reaction cross sections for the various angular momenta would normally be expected to vary randomly within a given range. This would result in a total cross section which is the product of the number of contributing partial waves times the average cross section within the range. For the specific reaction considered here, however, this averaging out does not occur, since the cross sections for the partial waves are nearly independent of the collisional angular momentum. This phenomenon is called phase locking. Phase locking has been predicted in recent years and has been subsequently experimentally observed to various degrees. The observation, discussed in the present work, is particularly clear. The authors claim that their results “constitute the first complete demonstration of partial-wave phase locking in a chemical reaction”. In order to show that the relaxation rate in the Rb / Rb⁺ collision is much smaller than the expected average, the authors compare the observed cross sections to spin changing collisions of Rb with a Sr⁺ ion where phase locking does not occur and averaging takes place.

The quality of the presented experimental and theoretical work is very high. The paper is well written. I think that it is generally worthy of being published in Nature Communications.

We thank the referee for their thoughtful and positive review. We appreciate their recognition of the quality of our experimental and theoretical work and their clear summary of the significance of our results.

I have a number of questions and comments which should be resolved before the work can be accepted: a) The authors state that the hyperfine deexcitation in the Rb-Rb⁺ collision happens in the course of charge exchange between the two atoms; i.e. where the valence electron hops between the two atoms. Besides the spin-exchange interaction, the Rb spin could in principle also flip due to spin- dipolar interaction. Can we exclude such other processes? For the relaxation in the collision with Sr⁺ the authors state that there can be variety of physical processes that can contribute to this effect.

We thank the referee for this important question. In the Rb-Rb⁺ system, to the best of our knowledge, charge exchange is the dominant mechanism, while other conceivable pathways are strongly suppressed.

A dipolar-driven flip of the Rb hyperfine state would require direct coupling to the *nuclear* spin of the Rb⁺ ion. The nuclear magnetic moment of ⁸⁷Rb is more than three orders of magnitude smaller than the magnetic moment of the electron. Even in systems like ⁸⁸Sr⁺-⁸⁷Rb, where dipolar interactions couple to an *electronic* spin, they contribute only a very small fraction of the hyperfine relaxation. Scaling this down by $\sim 10^{-3}$ for the nuclear spin suggests that dipolar relaxation in Rb-Rb⁺ is negligible. For completeness, bounding the dipole-dipole coupling as $V_{dd}(R) = \frac{\mu_0 \mu_B \mu_I}{4\pi R^3}$ at a very short interatomic separation $R = 5a_0$ gives $V_{dd} < 10^{-4} \text{cm}^{-1}$, which is at least six orders of magnitude smaller than the short-range exchange interaction in Rb (see, e.g. [PRA 106, 032804 (2022)]).

Spin-orbit and spin-rotation mechanisms are also expected to be extremely weak. Both require admixture of nonzero-L electronic character into the short-range molecular wavefunction. In Rb^+ the first excited manifold lies at approximately 16 eV above the ground state, so nonadiabatic mixing is strongly suppressed at all internuclear separations. Consistent with this, Rb_2^+ calculations [PRA 106, 032804 (2022)] that include relativistic effects (spin-orbit plus Breit) shift the relevant S-manifold energies by at most $< 1\text{cm}^{-1}$, which we take as an upper bound on the total spin-dependent scale in our entrance channel. By contrast, the short-range exchange splitting that drives resonant charge exchange reaches hundreds to thousands of cm^{-1} . This quantitative hierarchy, together with the absence of any near-degeneracies with higher-L curves in Rb-Rb^+ , implies that both spin-orbit and spin-rotation pathways are negligible relative to charge exchange.

In contrast, the exchange interaction between a closed-shell ion and its parent atom is intrinsically strong and, (absent phase locking) should yield large charge-exchange cross sections. In the resonant homonuclear case one expects a charge-exchange probability approaching one-half per Langevin encounter (see, e.g., Fig. 5b in [R. Côté, Adv. At. Mol. Opt. Phys. 65, 67 (2016)]).

In the revised manuscript we have added a Methods section that summarizes this analysis and refer the reader to this section in the main text when stating that charge-exchange is the dominant process (third paragraph under “Experimental Study” section).

b) The authors stress how important the calibration relative to the Langevin collision is. Why is it not enough to simply show (as done in Fig. 2a)) that the inelastic rate for the case of Rb^+ is much smaller than for Sr^+ ? Both rates are limited by the Langevin cross section, because the processes occur at close range. If one is much smaller than the other one, then this should be due to phase locking, right?

We thank the reviewer for this important question. We agree that both spin-exchange and charge-exchange processes occur at short range and therefore share essentially the same Langevin capture rate and probability in our two-ion geometry. However, because we do not directly measure the total charge-exchange rate, this comparison alone is insufficient. Our observable is hyperfine de-excitation resonant charge exchange (HDRCE), which registers only those charge-exchange events that are also accompanied by a hyperfine transition of the Rb atom. In other words, our measurement reflects the combined action of charge exchange and the hyperfine interaction.

Since HDRCE is a spin-dependent subset of all charge-exchange events, its rate includes spin-statistical branching factors that can reduce the measured signal even when the underlying charge-exchange probability is unsuppressed. Thus, a direct comparison of the raw $\text{Rb}^+\text{-Rb}$ and $\text{Sr}^+\text{-Rb}$ Langevin rates cannot by itself disentangle suppression due to spin-branching from true dynamical suppression of the charge-exchange process. A direct comparison with the spin-exchange rate of $\text{Sr}^+\text{-Rb}$ would similarly require separating the spin-dependent branching of the spin exchange interaction, which can differ from that of charge exchange.

To address this, we calibrate the HDRCE rate to the in-situ Langevin collision rate and use a theoretical multi-channel scattering model to explicitly bound the contribution of those spin-statistical prefactors. This procedure allows us to demonstrate that the observed suppression is far too large to be explained by spin-branching alone. Instead, it reflects a genuine dynamical suppression consistent with partial-wave phase locking.

To avoid confusion for future readers, we have added clarifying text to the revised manuscript (fifth paragraph under “Experimental Study”) explaining the rationale of these measurements.

c) The calibration of the Langevin cross section involves several semiclassical models where e.g. the trajectories of the ion through the gas of atoms are simulated. For this, an accurate treatment of all kinds of excess micromotion, of the trap induced bound state effects, and other physics is needed. The system is really quite complex. Given the systematic uncertainties in all of these models, I wonder to what extent all of this leads to a sizeable uncertainty of the calibration.

We appreciate the referee’s concern. Calibrating the Langevin cross section in an atom-ion system can, in principle, be sensitive to many experimental and modeling details. Our strategy has been to design the calibration so that it relies as little as possible on assumptions.

To achieve this, we perform the calibration in a classical regime by deliberately exciting micromotion to energies in the Kelvin range. At these energies, the particle trajectories can be accurately described using classical mechanics. All parameters required for the model at these energies, including trap frequencies and geometry, micromotion amplitude, and detection efficiencies, are measured independently and then held fixed during the analysis. While the fitting procedure formally includes two free parameters, the effective low-energy temperature T and the Langevin coefficient κ_L , the high-energy data are governed almost entirely by κ_L . This makes the calibration effectively a single-parameter fit.

We verify the robustness of the procedure by repeating the calibration under different experimental conditions. These tests include varying the micromotion settings and comparing results for single-ion and two-ion crystals. The extracted values of κ_L agree very well across these configurations. Trap-induced bound states are known to affect the reaction probabilities at low collision energies, and we explicitly account for their contribution when converting the calibrated capture rate into the HDRCE rate, following the procedures described in [Nat. Phys. 19, 1573 (2023)] and consistent with [PRL 130 143004 (2023)].

While it is never possible to rule out completely the presence of unknown effects, we find no inconsistencies across these cross-checks that would indicate systematic errors in the calibration. The observed suppression, which is more than an order of magnitude below the classical prediction for the HDRCE rate, is therefore robust to plausible systematics. Nevertheless, acknowledging the inherent complexity of the system, we have deliberately softened our language, referring to the result as a *signature* of partial-wave phase locking rather than a *definitive observation*. Future measurements that map HDRCE as a function of collision energy, for example by systematically varying the initial atom kinetic energy, would provide an even stronger test by directly tracing the predicted energy dependence and revealing the transition to the classical regime, where the HDRCE rate is expected to increase with energy. We now discuss such forward-looking experiments in the Discussion section.

d) In Fig. 2b) the probability P_b increases with the micromotion. Is this increase mainly explained by the fact that in a collision with higher energy collision there is a higher probability to impart a large enough energy so that the optical shelving transition does not work anymore, or is there also an additional effective increase in the collision cross section?

This interpretation is correct. The observed increase in P_b arises from larger momentum changes in the secular motion, as micromotion energy (in a direction transverse to the beam) is transferred into secular motion during collisions. This leads to failed shelving attempts into the D orbital and a higher probability

of detecting a bright ion. Our simulations assume an energy-independent Langevin capture rate through a single constant κ_L . (Strictly speaking, for the rate or probability to remain constant, the scattering cross section must decrease as $1/\sqrt{E}$).

To clarify this in the manuscript we have added the following text in the caption of Fig. 2b: “*The observed increase in P_b with micromotion arises from larger momentum transfer into secular motion during collisions, which can cause shelving failures into the D orbital and result in a higher probability of detecting a bright ion.*”. We also indicate explicitly in the methods that κ_L is taken as an energy-independent parameter.

e) According to equation (3) the rate constant converges towards $l/(2 * l + 1) k_L$. If I use $l = 3/2$, this results in $3/8 k_L$, not $3 / 16 k_L$, as stated in the text. What am I missing here?

We thank the referee for carefully checking this point and for spotting this important typo. The equation was missing a factor $1/2$ and should read $l/(4l+2)$, which leads to the correct factor of $3/16$. Intuitively, the term $l/(2l+1)$ accounts for the spin statistical prefactors, while the additional factor of $1/2$ comes from the averaging of the \sin^2 terms associated with the different partial waves (or equivalently, from the fact that the charge-exchange cross section is one-half of the Langevin value in the cold regime [R. Côté, Adv. At. Mol. Opt. Phys. 65, 67 (2016)]). The equation has been corrected in the revised text.

f) In Figure 3a) the experimental measurements are represented by a single dot at a particular temperature. How come in Figure 3b these experimental measurements are then represented by a black line over a large range temperature range?

We thank the referee for this important question. The experiment provides a single measurement at the mean collision temperature. In Fig. 3a, our aim is to display that data point and compare it against several theoretical models, highlighting which descriptions can reproduce the observed suppression and what this implies for the underlying short-range parameters.

By contrast, Fig. 3b is intended to place that single datum in a broader theoretical context. The colormap shows the predicted HDRCE rate as a function of temperature for one model, while the black line does not represent a scan of data but rather the locus of theory values consistent with the measured value of HDRCE suppression (with dashed white lines indicating the $\pm 1\sigma$ bounds). As the temperature increases and more partial waves contribute, phase locking weakens, the predicted rates rise toward the Langevin limit, and the constraint bends and eventually terminates where no parameter choice remains consistent with the measurement. The purpose of panel (b) is therefore to illustrate the temperature range over which the observed suppression is expected to disappear.

We have revised the caption of Fig. 3b and the text to make this distinction and the motivation explicit.

g) In the caption of Fig. 1 it reads “the contributions from different partial waves sum coherently and quantum interference can persist beyond the s-wave limit”. I am not happy how this sentence is formulated, as it is misleading. In quantum mechanics amplitudes might sum coherently, but not rates or cross sections. Quantum interference persists of course beyond the s-wave limit, but the effects of quantum interference might wash out.

We thank the referee and agree with their point. In response, we have revised the caption of Fig. 1 to read: “*At elevated temperature, many partial waves contribute to the reaction. In the classical regime,*

their contributions are summed incoherently, and thermal averaging tends to wash out quantum interference. If, however, the short-range phase difference between the gerade and ungerade channels changes only weakly with l and with collision energy, the interference factor remains nearly the same across many partial-wave terms. In this case, averaging over partial-wave contributions can still preserve wave-specific interference signatures beyond the s -wave limit, a phenomenon known as partial-wave phase locking [1].

h) In the “extended data Fig. 3”, the measured loss probabilities for Rb in the state $F = 1$ are still significantly lower for Rb+ than for Sr+. How can this be explained? From my understanding, the charge exchange suppression should not lead to any effects here.

We thank the referee for this question. In the $F=1$ manifold, hyperfine-changing transitions are suppressed, so no HDRCE signal is expected. The nonzero P_b observed in Extended Data Fig. 3 originates instead from rare background events in the logic-ion readout (occasional shelving failures due to a combination of finite ion temperature and a small per-logic-ion measurement error of $\sim 0.5\%$), rather than from reaction dynamics. This background scales with the number of logic ions: in Fig. 3 the purple bars correspond to data with two logic ions, while the gray bars correspond to a single logic ion, and the observed probability is approximately doubled as expected. Similar background levels were also observed in previous measurements with Sr^+ isotopes that are not optically addressed by our lasers [Nat. Phys. 18, 533 (2022)], supporting the conclusion that the $F=1$ signal arises from detection background rather than charge exchange processes.

To better clarify this point in the manuscript, we have added the following clarification in the paragraph presenting Extended Fig. 3: *“We therefore attribute this background to rare logic-ion readout errors (finite ion temperature and a $\sim 0.5\%$ per-logic-ion detection error), which scale with the number of logic ions measured and are independent of charge-exchange processes. Similar background levels were also observed in previous measurements with Sr^+ isotopes that are not optically addressed by our lasers [1].”*

i) In the discussion the authors write: “Trap-induced dynamics, while capable of enhancing reaction rates [48, 51], cannot account for the observed suppression and are explicitly included in our analysis.” This statement might be correct in the classical description, but would it also generally be correct in a quantum description where interference effects might take place?

We thank the referee for this comment. Our statement pertains to the temperature range of the present measurements, where many partial waves contribute and trap-induced dynamics are accurately described by classical trajectories. In this regime, trap-induced quasibound complexes increase the number of short-range returns and can enhance reaction rates, and we explicitly include these effects in our analysis. Since a fully quantum treatment of trap-assisted complexes in the ultracold limit lies beyond the scope of this work, we have revised the sentence in the Discussion to read: *“Trap-induced dynamics, which can enhance reaction rates by increasing the frequency of short-range encounters under classical motion [1], are explicitly included in our analysis.”*

j) In the outlook the authors write: “The techniques developed here, including in-situ Langevin rate calibration via momentum-changing collisions, open new opportunities for precision measurements in systems beyond the reach of current theoretical methods.” Can they explain more what they mean?

We thank the referee for this question. For many atom-ion systems, *ab initio* predictions of reaction cross sections remain out of reach due to the complexity of the short-range molecular potentials; for example, the resonant charge-exchange cross section of $\text{Cs}^+\text{-Cs}$ cannot currently be predicted reliably. The in-situ calibration and detection methods developed here provide a route to directly measure such reaction rate coefficients with high precision. The revised sentence now clarifies this point: *“The techniques developed in this work, including in-situ Langevin rate calibration via momentum-changing collisions, could open opportunities for precision measurements in systems where ab initio predictions remain out of reach, such as heavy atom-ion systems like $\text{Cs}^+\text{-Cs}$.”*

k) Reference [49] of the authors is not up-to-date, as it has been already published in *Science Advances*.

Done. We updated all preprints (including [49]) to their published version where applicable.

l) In the references there are repeated problems with missing capital letters.

Done. We thank the referee for spotting this Latex-bug and fixed it in the revised manuscript.